# META-LEARNING PRIORS USING UNROLLED PROXIMAL NETWORKS

**Yilang Zhang, Georgios B. Giannakis**
Department of Electric and Computer Engineering
University of Minnesota
Minneapolis, MN 55414, USA
`{zhan7453,georgios}@umn.edu`

## ABSTRACT

Relying on prior knowledge accumulated from related tasks, meta-learning offers a powerful approach to learning a novel task from limited training data. Recent approaches parameterize the prior with a family of probability density functions or recurrent neural networks, whose parameters can be optimized by utilizing validation data from the observed tasks. While these approaches have appealing empirical performance, the expressiveness of their prior is relatively low, which limits the generalization and interpretation of meta-learning. Aiming at expressive yet meaningful priors, this contribution puts forth a novel prior representation model that leverages the notion of algorithm unrolling. The key idea is to unroll the proximal gradient descent steps, where learnable piecewise linear functions are developed to approximate the desired proximal operators within *tight* theoretical error bounds established for both smooth and non-smooth proximal functions. The resultant multi-block neural network not only broadens the scope of learnable priors, but also enhances interpretability from an optimization viewpoint. Numerical tests conducted on few-shot learning datasets demonstrate markedly improved performance with flexible, visualizable, and understandable priors.

## 1 INTRODUCTION

While deep learning has achieved documented success in a broad range of applications (Silver et al., 2016; He et al., 2016; Vaswani et al., 2017), it often requires huge data records to train large-scale and high-capacity models. In contrast, human intelligence is capable of identifying new objects or concepts from merely a few samples. How to incorporate this ability into "machine intelligence" has garnered great attention and interest in a number of domains, especially when data are scarce or costly to collect. Examples of such applications include drug molecule discovery (Altae-Tran et al., 2017), low-resource machine translation (Gu et al., 2018), and robotics (Clavera et al., 2019).

Motivated by the fact that humans acquire new knowledge efficiently from past experiences, a principled framework has been investigated to mimic this ability of humans, known as *learning-to-learn* or *meta-learning* (Thrun & Pratt, 1998). Meta-learning aims to identify a task-invariant prior from a class of (partially) related tasks, which can be used to facilitate the learning of new tasks from the same class. The underlying assumption of meta-learning is that all tasks of interest are linked through their data distribution or latent problem structure. Thus, task-invariant common prior knowledge can be acquired as an inductive bias, and thereby transferred to new tasks (Thrun & Pratt, 1998). By doing so, even a couple of training data can suffice for learning a new task.

Conventional meta-learning methods rely on prescribed criteria to extract the prior; see e.g., (Schmidhuber, 1993; Bengio et al., 1995). With recent advances of deep learning, these hand-crafted approaches have been replaced by data-driven ones, where a meta-learner captures the prior information across tasks, while a base-learner utilizes this prior to aid per-task learning. The desired prior is encoded in the base-learner parameters shared across tasks, and can be learned by optimizing a loss over the given tasks. Early attempts to this end utilize a neural network (NN) to represent the prior (Santoro et al., 2016; Mishra et al., 2018; Ravi & Larochelle, 2017). The base-learner employs e.g., recurrent neural networks (RNNs) with input training data per task, and output parameters for

the task-specific model. However, the choices of the NNs heavily depend on the task-specific model, and the black-box nature of NNs makes them susceptible to poor interpretability and reliability.

As opposed to model-based meta-learning, model-agnostic meta-learning (MAML) extracts the prior without presuming the task-specific model beforehand (Finn et al., 2017). MAML resorts to an iterative optimizer to obtain the per-task model parameters. The prior information is reflected in the initialization of the model parameters, which is shared across tasks. Building upon MAML, various optimization-based meta-learning algorithms have been investigated to further improve its performance; see e.g., (Li et al., 2017; Bertinetto et al., 2019; Lee et al., 2019). Convergence guarantees have also been established to gain insights about these methods (Fallah et al., 2020; Ji et al., 2020; 2022). Interestingly, (Grant et al., 2018) pointed out that the initialization learned in MAML is approximately tantamount to the mean of a Gaussian prior probability density function (pdf) over the model parameters. This motivates well Bayesian formulations of meta-learning to further quantify the uncertainty in model parameters (Finn et al., 2018; Ravi & Beatson, 2019; Nguyen et al., 2020; Zhang et al., 2023). Nevertheless, the priors learned by these MAML-variants are confined to specific pdfs, including the Gaussian and degenerate ones. As a result, generalizing optimization-based meta-learning to practical domains that may require sophisticated priors is challenging.

This work advocates a novel meta-learning approach termed MetaProxNet that offers sufficient prior expressiveness, while maintaining the highly desirable interpretability. Our contribution is fourfold.

i) A prior representation framework is introduced using the algorithm unrolling technique. The novel framework overcomes the interpretability challenge and breaks the expressiveness bottleneck, thus enabling one to meta-learn complicated yet interpretable priors.

ii) Instead of employing a fixed proximal operator induced by a certain prior pdf, piecewise linear functions (PLFs) are developed to learn further generalized priors.

iii) Theoretical analysis provides tight PGD error bounds between the learnable PLFs and the optimal proximal operators, which can be readily minimized under mild conditions.

iv) Numerical tests compare MetaProxNet with state-of-the-art methods having different priors, and confirm superiority of MetaProxNet. PLFs are visualized to depict the explainable prior.

## 2 PROBLEM SETUP

Meta-learning extracts task-invariant prior information from a collection of relevant tasks to aid the learning of new tasks, even if only a small number of training data are available. Formally, let $t = 1, \ldots, T$ index the aforementioned relevant tasks, each with corresponding dataset $\mathcal{D}_t := \{(\mathbf{x}_t^n, y_t^n)\}_{n=1}^{N_t}$ comprising $N_t$ input-output data pairs. Set $\mathcal{D}_t$ is formed with a training subset $\mathcal{D}_t^{\mathrm{trn}} \subset \mathcal{D}_t$ and a validation subset $\mathcal{D}_t^{\mathrm{val}} := \mathcal{D}_t \setminus \mathcal{D}_t^{\mathrm{trn}}$. Likewise, a new task (with subscript $\star$) will comprise a training subset $\mathcal{D}_\star^{\mathrm{trn}}$, and a test input $\mathbf{x}_\star^{\mathrm{tst}}$, for which the corresponding output $y_\star^{\mathrm{tst}}$ is to be predicted. Typically, $|\mathcal{D}_\star^{\mathrm{trn}}|$ is rather small compared to what is required in supervised deep learning tasks. Due to the limited training data, directly learning the new task by optimizing its task-specific model over $\mathcal{D}_\star^{\mathrm{trn}}$ is infeasible. However, since $T$ can be considerably large, one prudent remedy is to leverage the cumulative prior knowledge across other related tasks.

Let $\boldsymbol{\theta}_t \in \mathbb{R}^d$ denote the task-specific model parameter for task $t$, and $\boldsymbol{\theta} \in \mathbb{R}^D$ the prior parameter shared across tasks. The prior can be learned via empirical risk minimization (ERM) *alternating* between i) base-learner optimization per $t$ that estimates $\boldsymbol{\theta}_t$ using $\mathcal{D}_t^{\mathrm{trn}}$ and $\boldsymbol{\theta}$; and, ii) meta-learner optimization that updates the estimate of $\boldsymbol{\theta}$ using $\{\mathcal{D}_t^{\mathrm{val}}\}_{t=1}^T$. This nested structure can be intrinsically characterized by a bilevel optimization problem

$$\min_{\boldsymbol{\theta}} \ \sum_{t=1}^T \mathcal{L}(\boldsymbol{\theta}_t^*(\boldsymbol{\theta}); \mathcal{D}_t^{\mathrm{val}}) \tag{1a}$$

$$\text{s.to} \quad \boldsymbol{\theta}_t^*(\boldsymbol{\theta}) = \operatorname*{argmin}_{\boldsymbol{\theta}_t} \mathcal{L}(\boldsymbol{\theta}_t; \mathcal{D}_t^{\mathrm{trn}}) + \mathcal{R}(\boldsymbol{\theta}_t; \boldsymbol{\theta}), \ \forall t \tag{1b}$$

where $\mathcal{L}$ is the loss function assessing the performance of the model, and $\mathcal{R}$ is the regularizer that captures the task-invariant prior. From the Bayesian viewpoint, $\mathcal{L}(\boldsymbol{\theta}_t; \mathcal{D}_t^{\mathrm{trn}})$ and $\mathcal{R}(\boldsymbol{\theta}_t; \boldsymbol{\theta})$ in (1b) are typically selected to be the negative log-likelihood (nll) $-\log p(\mathbf{y}_t^{\mathrm{trn}} | \boldsymbol{\theta}_t; \mathbf{X}_t^{\mathrm{trn}})$, and negative log-prior (nlp) $-\log p(\boldsymbol{\theta}_t; \boldsymbol{\theta})$, where matrix $\mathbf{X}_t^{\mathrm{trn}}$ is formed by all input vectors in $\mathcal{D}_t^{\mathrm{trn}}$, and $\mathbf{y}_t^{\mathrm{trn}}$ is the vector collecting their corresponding outputs. Hence, (1b) can be interpreted as the maximum a posteriori (MAP) estimator $\boldsymbol{\theta}_t^*(\boldsymbol{\theta}) = \operatorname*{argmax}_{\boldsymbol{\theta}_t} p(\boldsymbol{\theta}_t | \mathbf{y}_t^{\mathrm{trn}}; \mathbf{X}_t^{\mathrm{trn}}, \boldsymbol{\theta})$ upon invoking Bayes rule.

It is worth stressing that $\mathcal{R}(\boldsymbol{\theta}_t; \boldsymbol{\theta})$ is instrumental in learning task $t$, when $|\mathcal{D}_t^{\mathrm{trn}}|$ is small. Without it, an over-parameterized model such as a deep NN could easily overfit $\mathcal{D}_t^{\mathrm{trn}}$. Moreover, it is generally infeasible to reach the global minimum $\boldsymbol{\theta}_t^*$, especially with a highly non-convex optimization involved in learning the task-specific model. Thus, a practical alternative is to rely on a suboptimal solution $\hat{\boldsymbol{\theta}}_t$ obtained by a parameterized base-learner $\mathcal{B}$. Then, problem (1) boils down to

$$\min_{\boldsymbol{\theta}} \ \sum_{t=1}^{T} \mathcal{L}(\hat{\boldsymbol{\theta}}_t(\boldsymbol{\theta}); \mathcal{D}_t^{\mathrm{val}}) \tag{2a}$$

$$\text{s.to} \ \ \hat{\boldsymbol{\theta}}_t(\boldsymbol{\theta}) = \mathcal{B}(\mathcal{D}_t^{\mathrm{trn}}; \boldsymbol{\theta}), \ \forall t. \tag{2b}$$

Depending on the choices of $\mathcal{B}$, meta-learning approaches can be either NN-based or optimization-based ones. The former typically employ an RNN to learn the mapping from $\mathcal{D}_t^{\mathrm{trn}}$ to $\hat{\boldsymbol{\theta}}_t^*$, using the premise that the recurrent cells of an RNN correspond to the iterations for optimizing (1b) (Ravi & Larochelle, 2017). However, there is no analytical guarantee regarding the convergence of this "RNN-based optimization," and it is also hard to specify what priors have been learned by these RNNs. In contrast, the optimization-based approaches solve (1b) through an iterative optimizer, with $\mathcal{R}$ being the nlp term linked with a preselected pdf. For example, it has been reported in (Grant et al., 2018) that the optimization strategy adopted by MAML (Finn et al., 2017) corresponds up to an implicit Gaussian pdf $p(\boldsymbol{\theta}_t; \boldsymbol{\theta}) = \mathcal{N}(\boldsymbol{\theta}, \mathbf{Q}_t)$, where $\mathbf{Q}_t$ is associated with the hyperparameters of $\mathcal{B}$. Besides implicit prior pdfs, their explicit counterparts have also been investigated; see e.g., isotropic Gaussian (Rajeswaran et al., 2019), and diagonal Gaussian (Ravi & Beatson, 2019) examples.

## 3  Interpretable and generalized priors using unrolled NNs

Existing meta-learning algorithms rely on either a blackbox NN or a preselected pdf (such as a Gaussian one) to parameterize the prior. However, the NN often lacks interpretability and the chosen pdf can have limited expressiveness. Consider for instance a preselected Gaussian prior pdf, which is inherently unimodal, symmetric, log-concave, and infinitely differentiable by definition. Such a prior may not be well-suited for tasks with multimodal or asymmetric parametric pdfs; see App. I for a case study. To enhance the prior expressiveness as well as offer the desired interpretability, our key idea is to learn a *data-driven* regularizer $\mathcal{R}$, which dynamically adjusts its form to fit the provided tasks. This learnable $\mathcal{R}$ is effected by an unrolled NN, which drives our base-learner $\mathcal{B}$.

### 3.1  Prior representation via algorithm unrolling

Algorithm unrolling was introduced in (Gregor & LeCun, 2010) to learn the optimal update rule for the reconstruction of sparse signals from their low-dimensional linear measurements. In particular, algorithm unrolling involves unfolding the iterations of an optimization algorithm to create repeating blocks of an NN. In doing so, the desired prior is parameterized using learnable weights of the NN; see App J for a brief introduction. Following this work, several unrolling methods have been reported to learn interpretable priors for natural and medical signals, especially for images (Monga et al., 2021). Algorithm unrolling is also adopted here, but for a different purpose. While earlier efforts focus on learning the prior for a single task in the (transformed) *signal space* $\mathcal{X} \subseteq \mathbb{R}^{\dim(\mathbf{x}_t^n)}$, here it is employed for task-invariant prior extraction in the model *parameter space* $\boldsymbol{\Theta}_t \subseteq \mathbb{R}^d$; that is, the prior we aim to learn is $p(\boldsymbol{\theta}_t), \ \forall t$ rather than $p(\mathbf{x}_t^n)$ for $t$ given. The widely adopted convolutional (C)NNs, which exhibit remarkable effectiveness in representing priors for 2-dimensional images, may not fit well with the 1-dimensional $\boldsymbol{\theta}_t$. A better alternative will be sought after the ensuing discussion that links prior representation with proximal function learning.

To solve the regularized problem (1b), we consider unrolling the proximal gradient descent (PGD) algorithm (Parikh et al., 2014), which allows one to "divide and conquer" the objective function by separately optimizing $\mathcal{L}$ and $\mathcal{R}$. Each PGD iteration indexed by $k$ includes two steps: i) optimization of $\mathcal{L}(\boldsymbol{\theta}_t^{k-1}; \mathcal{D}_t^{\mathrm{trn}})$ wrt $\boldsymbol{\theta}_t^{k-1}$ using GD, with the update represented by an auxiliary variable $\mathbf{z}_t^k \in \mathbb{R}^d$; and ii) optimization of $\mathcal{R}(\boldsymbol{\theta}_t^{k-1}; \boldsymbol{\theta})$ using $\mathbf{z}_t^k$ to update $\boldsymbol{\theta}_t^{k-1}$. An upshot of the PGD algorithm is that it only requires $\mathcal{L}(\boldsymbol{\theta}_t; \cdot)$ to be differentiable wrt $\boldsymbol{\theta}_t$, while $\mathcal{R}(\boldsymbol{\theta}_t; \cdot)$ can be non-differentiable and even discontinuous. Thus, the expanded choices of $\mathcal{R}$ broaden the range of representable priors. The

---

**Algorithm 1:** Vanilla PGD algorithm for solving (1b)

**Input:** $\mathcal{D}_t^{\mathrm{trn}}$, hyperparameters $\boldsymbol{\theta}$, step size $\alpha$, and maximum iteration $K$.
**Initialization:** initialize $\boldsymbol{\theta}_t^0$ according to $\boldsymbol{\theta}$, and $\mathbf{z}_t^0 = \boldsymbol{\theta}_t^0$.

1 **for** $k = 1, \ldots, K$ **do**
2 $\quad$ Descend $\mathbf{z}_t^k = \boldsymbol{\theta}_t^{k-1} - \alpha \nabla_{\boldsymbol{\theta}_t^{k-1}} \mathcal{L}(\boldsymbol{\theta}_t^{k-1}; \mathcal{D}_t^{\mathrm{trn}})$;
3 $\quad$ Update $\boldsymbol{\theta}_t^k = \mathrm{prox}_{\mathcal{R}, \alpha}(\mathbf{z}_t^{k-1})$;
4 **end**

**Output:** $\hat{\boldsymbol{\theta}}_t = \boldsymbol{\theta}_t^K$.

---

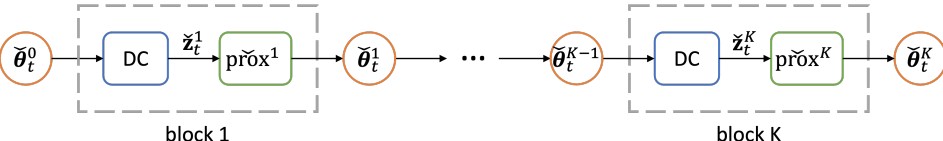

Figure 1: Illustrating diagram of the multi-block NN by unrolling the PGD algorithm.

steps of PGD are summarized in Algorithm 1, where the so-termed proximal operator is

$$\mathrm{prox}_{\mathcal{R}, \alpha}(\mathbf{z}) := \operatorname*{argmin}_{\boldsymbol{\theta}_t} \frac{1}{2\alpha} \|\boldsymbol{\theta}_t - \mathbf{z}\|_2^2 + \mathcal{R}(\boldsymbol{\theta}_t; \boldsymbol{\theta}). \tag{3}$$

For a broad range of $\mathcal{R}$, their corresponding $\mathrm{prox}_{\mathcal{R}, \alpha}$ has an analytical form. One well-known example is the indicator function $\mathcal{R} = \mathbb{I}_{\mathcal{S}}$ for some set $\mathcal{S}$, which is discontinuous and non-differentiable. However, it corresponds to a well-defined $\mathrm{prox}_{\mathcal{R}, \alpha}$, namely the projection operator $\mathbb{P}_{\mathcal{S}}$ onto set $\mathcal{S}$.

Using algorithm unrolling, our idea is to search for the unknown optimal regularizing function $\mathcal{R}^*$ (i.e., the one minimizing (1)) through learning its corresponding proximal operator $\mathrm{prox}_{\mathcal{R}^*, \alpha}$ with an unrolled NN. In particular, each PGD iteration indexed by $k$ is replaced by a block consisting of a data consistency (DC) module, and a learnable NN-based $\mathrm{p\check{r}ox}^k$. While the former ensures that the task-specific estimate $\check{\boldsymbol{\theta}}_t^{k-1}$ of the unrolled NN is consistent with $\mathcal{D}_t^{\mathrm{trn}}$ (by minimizing $\mathcal{L}(\check{\boldsymbol{\theta}}_t^{k-1}; \mathcal{D}_t^{\mathrm{trn}})$ wrt $\check{\boldsymbol{\theta}}_t^{k-1}$), the latter looks for the optimal per-step prior that calibrates $\check{\boldsymbol{\theta}}_t^{k-1}$. The pipeline of this unrolled NN is illustrated in Fig. 1, where the DC module can be either a naïve GD as in line 4 of Algorithm 1, or, a data-driven rule such as GD with a learnable $\alpha$. Let us for simplicity adopt the naïve GD as DC module, which aligns with MAML (Finn et al., 2017), and can be readily generalized to other iterative descent rules (Li et al., 2017; Lee & Choi, 2018; Park & Oliva, 2019; Flennerhag et al., 2020). The typical choice for each $\mathrm{p\check{r}ox}^k$ is an NN. Although $p(\boldsymbol{\theta}_t; \boldsymbol{\theta})$ may not be available since the NN mapping is nonlinear, it can serve as a generalized prior, if properly scaled.

Unlike previous works (Mardani et al., 2018; Hosseini et al., 2020) that model $\{\mathrm{p\check{r}ox}^k\}_{k=1}^K$ with 2-dimensional convolutions, here the input and output of $\mathrm{p\check{r}ox}^k$ are both 1-dimensional vectors in $\mathbb{R}^d$; cf. (3). Our motivation comes from the two most widely-used priors in optimization-based meta-learning. The first prior is the diagonal Gaussian one with $\mathcal{R}(\boldsymbol{\theta}_t; \boldsymbol{\theta}) = \frac{1}{2}(\boldsymbol{\theta}_t - \boldsymbol{\theta}^{\mathrm{init}})^\top \mathrm{diag}(\boldsymbol{\lambda})(\boldsymbol{\theta}_t - \boldsymbol{\theta}^{\mathrm{init}})$, where $\boldsymbol{\theta}^{\mathrm{init}} = \boldsymbol{\theta}_t^0$ is the task-invariant initialization of (1b), and $\boldsymbol{\theta} := [\boldsymbol{\theta}^{\mathrm{init}\top}, \boldsymbol{\lambda}^\top]^\top$ is the vector parameterizing $\mathcal{R}$ (Ravi & Beatson, 2019; Rajeswaran et al., 2019; Nguyen et al., 2020). It can be easily verified that $\mathrm{prox}_{\mathcal{R}, \alpha}(\mathbf{z}) = (\mathbf{z} - \boldsymbol{\theta}^{\mathrm{init}})/(\mathbf{1}_d + \alpha \boldsymbol{\lambda}) + \boldsymbol{\theta}^{\mathrm{init}}$, with / being the element-wise division and $\mathbf{1}_d \in \mathbb{R}^d$ denoting the constant vector of all 1's. The second example is the shifted sparse prior that shares a pre-defined portion of $\boldsymbol{\theta}_t$ across tasks (Raghu et al., 2020; Bertinetto et al., 2019; Lee et al., 2019). Here, we consider its variant $\mathcal{R}(\boldsymbol{\theta}_t; \boldsymbol{\theta}) = \|\boldsymbol{\Lambda}(\boldsymbol{\theta}_t - \boldsymbol{\theta}^{\mathrm{init}})\|_1$ that can be learned (Tian et al., 2020b). This results in $\mathrm{prox}_{\mathcal{R}, \alpha}(\mathbf{z}) = \mathbb{S}_{\alpha \boldsymbol{\lambda}}(\mathbf{z} - \boldsymbol{\theta}^{\mathrm{init}}) + \boldsymbol{\theta}^{\mathrm{init}}$, where $\mathbb{S}_{\alpha \boldsymbol{\lambda}}$ is the element-wise shrinkage (a.k.a. soft-thresholding) operator such that its $i$-th element

$$[\mathbb{S}_{\alpha \boldsymbol{\lambda}}(\mathbf{z})]_i := \mathbb{S}_{\alpha \lambda_i}(z_i) := \begin{cases} z_i + \alpha \lambda_i, & z_i < -\alpha \lambda_i \\ 0, & -\alpha \lambda_i \le z_i < \alpha \lambda_i \\ z_i - \alpha \lambda_i, & z_i \ge \alpha \lambda_i \end{cases}.$$

For notational simplicity, denote by shifted vectors $\bar{\boldsymbol{\theta}}_t^k := \boldsymbol{\theta}_t^k - \boldsymbol{\theta}^{\mathrm{init}}$, $\bar{\mathbf{z}}_t^k := \mathbf{z}_t^k - \boldsymbol{\theta}^{\mathrm{init}}$, shifted loss $\bar{\mathcal{L}}(\boldsymbol{\theta}; \cdot) := \mathcal{L}(\boldsymbol{\theta} + \boldsymbol{\theta}^{\mathrm{init}}; \cdot)$, and shifted proximal operator $\overline{\mathrm{prox}}_{\mathcal{R}, \alpha}(\mathbf{z}) := \mathrm{prox}_{\mathcal{R}, \alpha}(\mathbf{z} + \boldsymbol{\theta}^{\mathrm{init}}) - \boldsymbol{\theta}^{\mathrm{init}}$.

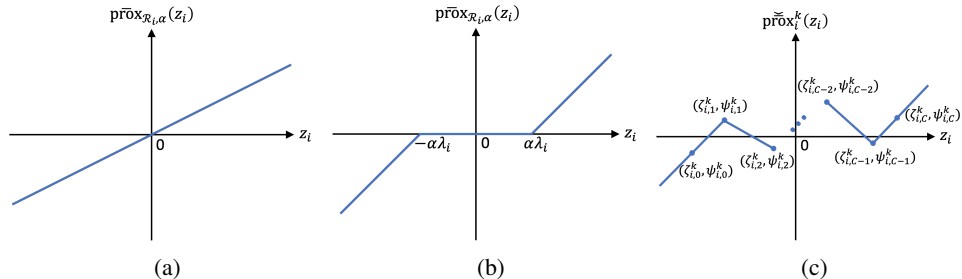

Figure 2: Proximal operators for: (a) Gaussian prior; (b) sparse prior; (c) unrolling-based prior.

The PGD iteration can be thus reformulated as

$$\bar{\mathbf{z}}_t^k = \bar{\boldsymbol{\theta}}_t^{k-1} - \alpha \nabla_{\bar{\boldsymbol{\theta}}_t^{k-1}} \bar{\mathcal{L}}(\bar{\boldsymbol{\theta}}_t^{k-1}; \mathcal{D}_t^{\mathrm{trn}}) \tag{4a}$$

$$\bar{\boldsymbol{\theta}}_t^k = \mathrm{p\bar{r}ox}_{\mathcal{R},\alpha}(\bar{\mathbf{z}}_t^k), \quad k = 1, \dots, K \tag{4b}$$

with initialization $\bar{\boldsymbol{\theta}}_t^0 = \bar{\mathbf{z}}_t^0 = \mathbf{0}_d$ and output $\hat{\boldsymbol{\theta}}_t = \bar{\boldsymbol{\theta}}_t^K + \boldsymbol{\theta}^{\mathrm{init}}$. Further, the $\mathrm{p\bar{r}ox}_{\mathcal{R},\alpha}(\mathbf{z})$ operator of the forgoing two examples reduces to $\mathbf{z}/(\mathbf{1}_d + \alpha\boldsymbol{\lambda})$ and $\mathbb{S}_{\alpha\boldsymbol{\lambda}}(\mathbf{z})$, respectively.

Inspired by the fact that $\mathrm{p\bar{r}ox}_{\mathcal{R},\alpha}(\mathbf{z})$ of both examples belongs to the family of piecewise linear functions (PLFs), the fresh idea is to parameterize the shifted per-step $\mathrm{p\check{r}ox}^k(\mathbf{z};\boldsymbol{\theta}) := \mathrm{p\check{r}ox}^k(\mathbf{z} + \boldsymbol{\theta}^{\mathrm{init}}) - \boldsymbol{\theta}^{\mathrm{init}}$ of the unrolled NN using learnable PLFs. We first show that the wanted $\mathrm{p\check{r}ox}^k : \mathbb{R}^d \mapsto \mathbb{R}^d$ can be effectively decomposed and thus simplified under the following assumption that is widely adopted in meta-learning (Ravi & Beatson, 2019; Rajeswaran et al., 2019; Nguyen et al., 2020).

**Assumption 3.1.** The optimal regularizer $\mathcal{R}^*$ factorizes across its input dimensions; that is, $\mathcal{R}^*(\boldsymbol{\theta}_t;\boldsymbol{\theta}) = \sum_{i=1}^d \mathcal{R}_i^*([\boldsymbol{\theta}_t]_i;\boldsymbol{\theta})$.

With Assumption 3.1 in effect, an immediate result is the element-wise proximal operator

$$[\mathrm{prox}_{\mathcal{R}^*,\alpha}(\mathbf{z})]_i = \underset{[\boldsymbol{\theta}_t]_i}{\mathrm{argmin}} \; \frac{1}{2\alpha} \|\boldsymbol{\theta}_t - \mathbf{z}\|_2^2 + \sum_{i=1}^d \mathcal{R}_i^*([\boldsymbol{\theta}_t]_i;\boldsymbol{\theta})$$

$$= \underset{[\boldsymbol{\theta}_t]_i}{\mathrm{argmin}} \; \frac{1}{2\alpha} ([\boldsymbol{\theta}_t]_i - z_i)^2 + \mathcal{R}_i^*([\boldsymbol{\theta}_t]_i;\boldsymbol{\theta}) := \mathrm{prox}_{\mathcal{R}_i^*,\alpha}(z_i), \quad i = 1, \dots, d. \tag{5}$$

This observation suggests that we can alternatively model the dimension-wise decomposition $\mathrm{p\check{r}ox}_i^k := [\mathrm{p\check{r}ox}^k]_i$ for each $i = 1, \dots, d$, with a handy 1-dimensional PLF

$$\mathrm{p\check{r}ox}_i^k(z_i) = \begin{cases} \frac{\psi_{i,0}^k(\zeta_{i,1}^k - z_i) + \psi_{i,1}^k(z_i - \zeta_{i,0}^k)}{\zeta_{i,1}^k - \zeta_{i,0}^k}, & z_i < \zeta_{i,1}^k \\ \frac{\psi_{i,c-1}^k(\zeta_{i,c}^k - z_i) + \psi_{i,c}^k(z_i - \zeta_{i,c-1}^k)}{\zeta_{i,c}^k - \zeta_{i,c-1}^k}, & \zeta_{i,c-1}^k \leq z_i < \zeta_{i,c}^k \\ & \text{and } c = 2, \dots C - 1 \\ \frac{\psi_{i,C}^k(\zeta_{i,C+1}^k - z_i) + \psi_{i,C+1}^k(z_i - \zeta_{i,C}^k)}{\zeta_{i,C+1}^k - \zeta_{i,C}^k}, & z_i \geq \zeta_{i,C-1}^k \end{cases} \tag{6}$$

where $C \geq 1$ is a pre-selected constant indicating the total number of pieces, and $\{(\zeta_{i,c}^k, \psi_{i,c}^k)\}_{c=0}^C$ are the learnable control points parametrizing $\mathrm{p\check{r}ox}_i^k$. To ensure $\mathrm{p\check{r}ox}_i^k$ is a valid function, we further require $\zeta_{i,0}^k \leq \dots \leq \zeta_{i,C}^k$ for $\forall i, k$. To this end, the problem of finding a proper task-invariant prior $p(\boldsymbol{\theta}_t;\boldsymbol{\theta})$ boils down to learning the parameters of PLFs that are shared across tasks. Comparison of the pdf-based and PLF-based proximal operators can be visualized in Fig. 2.

## 3.2 PRIOR LEARNING VIA ALTERNATING OPTIMIZATION

Building upon the unrolling-based prior information representation, we are ready to elucidate how the prior can be learned by alternately optimizing the meta-learner and base-learner. We term the proposed method as meta-learning via proximal networks (MetaProxNet).

Let $r$ and $k$ denote iteration indices for (1a) and (1b), respectively. For notational brevity, define vectors $\boldsymbol{\zeta}^k := [\zeta_{1,0}^k, \dots, \zeta_{d,C}^k]^\top$ and $\boldsymbol{\psi}^k := [\psi_{1,0}^k, \dots, \psi_{d,C}^k]^\top$ of the PLF control points, and $\boldsymbol{\theta}^r$ the concatenation of $\boldsymbol{\theta}^{\mathrm{init},r}, \boldsymbol{\zeta}^{1,r}, \dots, \boldsymbol{\zeta}^{K,r}, \boldsymbol{\psi}^{1,r}, \dots, \boldsymbol{\psi}^{K,r}$ in the $r$-th iteration of (1a). Given

---

**Algorithm 2:** MetaProxNet algorithm

---

**Input:** $\{\mathcal{D}_t\}_{t=1}^T$, step sizes $\alpha$ and $\beta$, batch size $B$, and maximum iterations $K$ and $R$.
**Initialization:** randomly initialize $\boldsymbol{\theta}^0$.

1 **for** $r = 1, \ldots, R$ **do**
2     Randomly sample a mini-batch $\mathcal{T}^r \subset \{1, \ldots, T\}$ of cardinality $B$;
3     **for** $t \in \mathcal{T}^r$ **do**
4        Initialize $\breve{\bar{\boldsymbol{\theta}}}_t^0 = \breve{\mathbf{z}}_t^0 = \mathbf{0}_d$;
5        **for** $k = 1, \ldots, K$ **do**
6           Descend $\breve{\mathbf{z}}_t^k(\boldsymbol{\theta}^{r-1}) = \breve{\bar{\boldsymbol{\theta}}}_t^{k-1}(\boldsymbol{\theta}^{r-1}) - \alpha\nabla_{\breve{\bar{\boldsymbol{\theta}}}_t^{k-1}}\bar{\mathcal{L}}(\breve{\bar{\boldsymbol{\theta}}}_t^{k-1}(\boldsymbol{\theta}^{r-1}); \mathcal{D}_t^{\mathrm{trn}})$;
7           Update $\breve{\bar{\boldsymbol{\theta}}}_t^k(\boldsymbol{\theta}^{r-1}) = \breve{\mathrm{prox}}^k(\breve{\mathbf{z}}_t^k(\boldsymbol{\theta}^{r-1}); \boldsymbol{\zeta}^{k,r}, \boldsymbol{\psi}^{k,r})$;
8        **end**
9        Shift $\hat{\boldsymbol{\theta}}_t(\boldsymbol{\theta}^{r-1}) = \breve{\bar{\boldsymbol{\theta}}}_t^K(\boldsymbol{\theta}^{r-1}) + \boldsymbol{\theta}^{\mathrm{init},r}$;
10     **end**
11     Update $\boldsymbol{\theta}^r = \boldsymbol{\theta}^{r-1} - \beta\frac{1}{B}\sum_{t\in\mathcal{T}^r}\nabla_{\boldsymbol{\theta}^{r-1}}\mathcal{L}(\hat{\boldsymbol{\theta}}_t(\boldsymbol{\theta}^{r-1}); \mathcal{D}_t^{\mathrm{val}})$;
12 **end**

**Output:** $\hat{\boldsymbol{\theta}} = \boldsymbol{\theta}^R$.

---

$\{\mathcal{D}_t^{\mathrm{trn}}\}_{t=1}^T$, the goal of (1b) is to learn the task-specific estimate $\hat{\boldsymbol{\theta}}_t(\boldsymbol{\theta}^r)$ that depends on $\boldsymbol{\theta}^r$ per task $t$. This can leverage the current base-learner estimate $\mathcal{B}(\cdot; \boldsymbol{\theta}^r)$, which is the unrolled multi-block NN of our MetaProxNet. In the $k$-th block, its DC module and PLFs optimize (1b) through

$$\breve{\mathbf{z}}_t^k(\boldsymbol{\theta}^r) = \breve{\bar{\boldsymbol{\theta}}}_t^{k-1}(\boldsymbol{\theta}^r) - \alpha\nabla_{\breve{\bar{\boldsymbol{\theta}}}^{k-1}}\bar{\mathcal{L}}(\breve{\bar{\boldsymbol{\theta}}}_t^{k-1}(\boldsymbol{\theta}^r); \mathcal{D}_t^{\mathrm{trn}}) \tag{7a}$$

$$\breve{\bar{\boldsymbol{\theta}}}_t^k(\boldsymbol{\theta}^r) = \breve{\mathrm{prox}}^k(\breve{\mathbf{z}}_t^k(\boldsymbol{\theta}^r); \boldsymbol{\zeta}^{k,r}, \boldsymbol{\psi}^{k,r}), k = 1, \ldots, K. \tag{7b}$$

where $\breve{\mathbf{z}}_t^k$ and $\breve{\bar{\boldsymbol{\theta}}}_t^k$ denote the shifted iterative variables of the unrolled NN as in (4).

After obtaining $\hat{\boldsymbol{\theta}}_t(\boldsymbol{\theta}^r) = \breve{\bar{\boldsymbol{\theta}}}_t^K(\boldsymbol{\theta}^r) + \boldsymbol{\theta}^{\mathrm{init},r}$, the next step is to optimize (1a) by updating $\boldsymbol{\theta}^r$. A popular strategy is the mini-batch stochastic GD (SGD). Specifically, a subset $\mathcal{T}^r \subset \{1, \ldots, T\}$ of tasks are randomly selected to assess the performance of $\boldsymbol{\theta}^r$ on $\mathcal{D}_t^{\mathrm{val}}$, which yields a loss $\mathcal{L}(\hat{\boldsymbol{\theta}}_t(\boldsymbol{\theta}^r); \mathcal{D}_t^{\mathrm{val}})$ for $\forall t \in \mathcal{T}^r$. Then, $\boldsymbol{\theta}^{r+1}$ is reached by descending the averaged loss with step size $\beta$, that is

$$\boldsymbol{\theta}^{r+1} = \boldsymbol{\theta}^r - \beta\frac{1}{|\mathcal{T}^r|}\sum_{t\in\mathcal{T}^r}\nabla_{\boldsymbol{\theta}^r}\mathcal{L}(\hat{\boldsymbol{\theta}}_t(\boldsymbol{\theta}^r); \mathcal{D}_t^{\mathrm{val}}). \tag{8}$$

The step-by-step pseudo-codes for our novel MetaProxNet approach are listed under Algorithm 2.

In practice however, simultaneously optimizing both $\{\boldsymbol{\zeta}^k\}_{k=1}^K$ and $\{\boldsymbol{\psi}^k\}_{k=1}^K$ incurs cumbersome gradient computations due to the entangled structure of (1). To relieve this burden, we fix the former by uniformly partitioning a closed interval $[-A, A]$, while optimizing only the latter. In other words, we let $\zeta_{i,c}^k = (\frac{2c}{C} - 1)A$, $\forall c, i, k$, where $A > 0$ is a pre-selected constant that is sufficiently large; see Assumption A.3. In fact, this setup can be viewed as a uniform discretization of the continuous variable $\zeta_i^k \in \mathbb{R}$ on $[-A, A]$. Non-uniform discretization can be alternatively sought, if $p(\zeta_i^k)$ or its estimate is available a priori.

### 3.3 ERROR BOUNDS FOR PLF-BASED PROXIMAL OPERATOR

Having introduced how to model and learn priors using unrolled NNs, this subsection analyzes the performance by bounding the approximation error on $\hat{\boldsymbol{\theta}}_t$ induced by replacing the unknown optimal $\overline{\mathrm{prox}}_{\mathcal{R}^*,\alpha}$ with the learned PLF-based $\breve{\mathrm{prox}}^k$. Sharp bounds will be separately established for smooth and non-smooth $\overline{\mathrm{prox}}_{\mathcal{R}^*,\alpha}$ operators under mild conditions. Utilizing these bounds, a quantitative criterion will be provided for choosing the hyperparameter $C$. All proofs and technical assumptions can be found in Apps. A-C. Smooth $\overline{\mathrm{prox}}_{\mathcal{R}^*,\alpha} \in \mathcal{C}^1([-A, A]^d)$ will be first considered.

The following theorem offers an upper bound for the normalized error on (shifted) $\hat{\boldsymbol{\theta}}_t$.

**Theorem 3.2** (Finite-step PGD error for smooth proximal operators)**.** *Consider* $\breve{\mathrm{prox}}^k$ *defined by (6) with fixed* $\zeta_{i,c}^k = (\frac{2c}{C} - 1)A$, *and let* $\boldsymbol{\Psi} := [\boldsymbol{\psi}^1, \ldots, \boldsymbol{\psi}^K]$ *denote the matrix parameterizing* $\{\breve{\mathrm{prox}}^k\}_{k=1}^K$. *Let* $\bar{\boldsymbol{\theta}}_t^K$ *and* $\breve{\bar{\boldsymbol{\theta}}}_t^K$ *be the $K$-step PGD outputs using* $\overline{\mathrm{prox}}_{\mathcal{R}^*,\alpha} \in \mathcal{C}^1([-A, A]^d)$ *and*

p̌rox$^k$, *respectively. Under mild assumptions, it holds for $t = 1, \ldots, T$ that*

$$\min_{\boldsymbol{\Psi}} \frac{1}{\sqrt{d}} \big\| \bar{\boldsymbol{\theta}}_t^K - \check{\boldsymbol{\theta}}_t^K(\boldsymbol{\Psi}) \big\|_2 = \mathcal{O}(\frac{1}{C^2}). \tag{9}$$

*This bound is tight when $\psi_{i,0}^k = \bar{\text{prox}}_{\mathcal{R}_i^*,\alpha}(-A)$ and $\psi_{i,C}^k = \bar{\text{prox}}_{\mathcal{R}_i^*,\alpha}(A)$, $\forall k, i$.*

Theorem 3.2 asserts that by optimizing over $\boldsymbol{\Psi}$ of the PLFs, p̌rox$^k$ can approximate *any* smooth $\bar{\text{prox}}_{\mathcal{R}^*,\alpha}$ with $K$-step PGD error in the order $\mathcal{O}(\frac{1}{C^2})$. In other words, an $\epsilon$-approximant $\check{\boldsymbol{\theta}}_t^K$ of $\bar{\boldsymbol{\theta}}_t^K$ can be obtained upon choosing $C = \Omega(\frac{1}{\sqrt{\epsilon}})$ and optimizing $\boldsymbol{\Psi}$. The tightness of the bound implies that there exists at least one $\bar{\text{prox}}_{\mathcal{R}^*,\alpha}$ that reaches the upper bound when enforcing the first and last control points of each PLF to align with the desired $\bar{\text{prox}}_{\mathcal{R}_i^*,\alpha}$ operator.

Unfortunately, directly optimizing the left-hand side of (9) is impossible, because the optimal $\bar{\text{prox}}_{\mathcal{R}^*,\alpha}$ corresponding to the oracle prior $p(\boldsymbol{\theta}_t; \boldsymbol{\theta}^*)$ is unknown. A feasible alternative is to perform the ERM in (1) by leveraging the datasets $\{\mathcal{D}_t\}_{t=1}^T$ generated with $\boldsymbol{\theta}_t \sim p(\boldsymbol{\theta}_t; \boldsymbol{\theta}^*)$. As a result, the (unknown) optimal PLF parameters $\boldsymbol{\Psi}^* = \arg\min_{\boldsymbol{\Psi}} \big\| \bar{\boldsymbol{\theta}}_t^K - \check{\boldsymbol{\theta}}_t^K(\boldsymbol{\Psi}) \big\|_2$, and the sub-optimal estimate $\hat{\boldsymbol{\Psi}}$ obtained by solving (1), satisfy the inequality

$$\frac{1}{\sqrt{d}} \big\| \bar{\boldsymbol{\theta}}_t^K - \check{\boldsymbol{\theta}}_t^K(\hat{\boldsymbol{\Psi}}) \big\|_2 \le \frac{1}{\sqrt{d}} \big\| \bar{\boldsymbol{\theta}}_t^K - \check{\boldsymbol{\theta}}_t^K(\boldsymbol{\Psi}^*) \big\|_2 + \frac{1}{\sqrt{d}} \big\| \check{\boldsymbol{\theta}}_t^K(\hat{\boldsymbol{\Psi}}) - \check{\boldsymbol{\theta}}_t^K(\boldsymbol{\Psi}^*) \big\|_2. \tag{10}$$

The extra error $\frac{1}{\sqrt{d}} \big\| \check{\boldsymbol{\theta}}_t^K(\hat{\boldsymbol{\Psi}}) - \check{\boldsymbol{\theta}}_t^K(\boldsymbol{\Psi}^*) \big\|_2$ can be further bounded in linear order $\mathcal{O}(\frac{1}{\sqrt{d}} \| \hat{\boldsymbol{\Psi}} - \boldsymbol{\Psi}^* \|_1)$ of the normalized ERM error; see App. C for further elaboration.

Aside from smooth ones, non-smooth $\bar{\text{prox}}_{\mathcal{R}^*,\alpha}$ has gained attention in various PGD-guided applications. The next theorem forgoes the smooth assumption to yield a more generic but looser bound.

**Theorem 3.3** (Finite-step PGD error for continuous proximal operators). *Consider the notational conventions of Theorem 3.2 with continuous $\bar{\text{prox}}_{\mathcal{R}^*,\alpha} \in \mathcal{C}^0([-A, A]^d)$. Under mild assumptions, it holds for $t = 1, \ldots, T$ that*

$$\min_{\boldsymbol{\Psi}} \frac{1}{\sqrt{d}} \big\| \bar{\boldsymbol{\theta}}_t^K - \check{\boldsymbol{\theta}}_t^K(\boldsymbol{\Psi}) \big\|_2 = \mathcal{O}(\frac{1}{C}). \tag{11}$$

*This bound is tight when $\psi_{i,0}^k = \bar{\text{prox}}_{\mathcal{R}_i^*,\alpha}(-A)$ and $\psi_{i,C}^k = \bar{\text{prox}}_{\mathcal{R}_i^*,\alpha}(A)$, $\forall k, i$.*

Compared to the smooth case, the error bound in Theorem 3.3 has an order of $\mathcal{O}(\frac{1}{C})$. This implies that by selecting $C = \Omega(\frac{1}{\epsilon})$, operator p̌rox$^k$ can approximate *any* continuous $\bar{\text{prox}}_{\mathcal{R}^*,\alpha}$ with normalized $K$-step PGD error no larger than $\epsilon$. This increased order implies that one can easily expand the range of learnable priors with a larger $C$. Moreover, the discussion following (10) regarding the sub-optimality of $\hat{\boldsymbol{\Psi}}$, applies to Theorem 3.3 too, and it is deferred to App. C.

## 4 NUMERICAL TESTS

In this section, numerical tests are presented on several meta-learning benchmark datasets to evaluate the empirical performance of MetaProxNet. Hyperparameters and datasets are described in App. E. All experiments are run on a server with RTX A5000 GPU, and our codes are available online at `https://github.com/zhangyilang/MetaProxNet`.

### 4.1 COMPARISON OF META-LEARNING METHODS HAVING DIFFERENT PRIORS

The first test is on few-shot classification datasets miniImageNet (Vinyals et al., 2016) and Tiered-ImageNet (Ren et al., 2018), where "shot" signifies the per-class number of labeled training data for each $t$. The default model is a standard 4-layer CNN (Vinyals et al., 2016), each layer comprising a $3 \times 3$ convolution operation of 64 channels, a batch normalization, a ReLU activation, and a $2 \times 2$ max pooling. A linear regressor with softmax is appended to perform classification.

To demonstrate the superiority of unrolling-based priors over the RNN-based and handcrafted ones, we first compare MetaProxNet against several state-of-the-art meta-learning methods. As discussed in Sec. 3.1, our MetaProxNet can be readily integrated with other optimization-based meta-learning

Table 1: Comparison of MetaProxNet against meta-learning methods with different priors. The highest accuracy as well as mean accuracies within its $95\%$ confidence interval are bolded.

| Method | Prior | 5-class miniImageNet | | 5-class TieredImageNet | |
|---|---|---|---|---|---|
| | | 1-shot (%) | 5-shot (%) | 1-shot (%) | 5-shot (%) |
| LSTM (Ravi & Larochelle, 2017) | RNN-based | $43.44_{\pm0.77}$ | $60.60_{\pm0.71}$ | $-$ | $-$ |
| MAML (Finn et al., 2017) | implicit Gaussian | $48.70_{\pm1.84}$ | $63.11_{\pm0.92}$ | $51.67_{\pm1.81}$ | $70.30_{\pm1.75}$ |
| ProtoNets (Snell et al., 2017) | shifted sparse | $49.42_{\pm0.78}$ | $68.20_{\pm0.66}$ | $53.31_{\pm0.87}$ | $72.69_{\pm0.74}$ |
| R2D2 (Bertinetto et al., 2019) | shifted sparse | $51.8_{\pm0.2}$ | $68.4_{\pm0.2}$ | $-$ | $-$ |
| MC (Park & Oliva, 2019) | block-diag. Gaussian | $54.08_{\pm0.93}$ | $67.99_{\pm0.73}$ | $-$ | $-$ |
| L2F (Baik et al., 2020) | implicit Gaussian | $52.10_{\pm0.50}$ | $69.38_{\pm0.46}$ | $54.40_{\pm0.50}$ | $\mathbf{73.34_{\pm0.44}}$ |
| KML (Abdollahzadeh et al., 2021) | shifted sparse | $54.10_{\pm0.61}$ | $68.07_{\pm0.45}$ | $54.67_{\pm0.39}$ | $72.09_{\pm0.27}$ |
| MeTAL (Baik et al., 2021) | implicit Gaussian | $52.63_{\pm0.37}$ | $70.52_{\pm0.29}$ | $54.34_{\pm0.31}$ | $70.40_{\pm0.21}$ |
| MinimaxMAML (Wang et al., 2023) | inverted nlp | $51.70_{\pm0.42}$ | $68.41_{\pm1.28}$ | $-$ | $-$ |
| MetaProxNet+MAML | unrolling-based | $53.70_{\pm1.40}$ | $70.08_{\pm0.69}$ | $54.56_{\pm1.44}$ | $71.80_{\pm0.73}$ |
| MetaProxNet+MC | unrolling-based | $\mathbf{55.94_{\pm1.39}}$ | $\mathbf{71.97_{\pm0.67}}$ | $\mathbf{57.34_{\pm1.42}}$ | $\mathbf{73.38_{\pm0.73}}$ |

Table 2: Ablation tests of MetaProxNet using miniImageNet dataset with a 4-layer 32-channel CNN.

| Method | Preset prior? | DC | 5-class | | 10-class | |
|---|---|---|---|---|---|---|
| | | | 1-shot (%) | 5-shot (%) | 1-shot (%) | 5-shot (%) |
| MAML | Yes | GD | $48.70_{\pm1.84}$ | $63.11_{\pm0.92}$ | $31.27_{\pm1.15}$ | $46.92_{\pm1.25}$ |
| PGD-Gaussian | Yes | PGD | $48.58_{\pm1.40}$ | $64.56_{\pm0.70}$ | $30.04_{\pm0.83}$ | $47.30_{\pm0.49}$ |
| MetaProxNet+MAML | No | PGD | $\mathbf{53.58_{\pm1.43}}$ | $\mathbf{67.88_{\pm0.72}}$ | $\mathbf{34.80_{\pm0.91}}$ | $\mathbf{51.03_{\pm0.51}}$ |

methods through a simple substitution of the DC module. Tab. 1 lists the performance of MetaProx-Net assessed using $1,000$ random new tasks, with MAML (Finn et al., 2017) and MetaCurvature (MC) (Park & Oliva, 2019) serving as backbones. For an apples-to-apples comparison, methods that use different models (e.g., residual networks) or pretrained feature extractors are not included in the table. It is seen that our MetaProxNet performs competitively in terms of classification accuracy when compared to state-of-the-art meta-learning methods. This empirically confirms the effectiveness of MetaProxNet. Additional discussions regarding the efficiency of MetaProxNet and extra tests with tied weights can be found in the Apps. F and G.

## 4.2 ABLATION TESTS

Ablation tests are also carried out to investigate the essential reason for the performance gain of MetaProxNet. Evidently, MetaProxNet+MAML differs from its backbone MAML in two key aspects: task-level optimization algorithm (PGD vs. GD) and prior (unrolled-NN based vs. Gaussian). To assess which of the two contributes more to the performance gain of MetaProxNet, the ablation tests compare three methods: i) MAML that employs GD and Gaussian prior; ii) a variant with PGD and Gaussian prior; and, iii) MetaProxNet+MAML that utilizes PGD and an unrolled-NN based prior. To avoid overfitting in MAML, the models for all methods are fixed to a 4-layer 32-channel CNN. Tab. 2 lists the performance of the three methods. It is seen that the PGD baseline and MAML exhibit comparable performance, while MetaProxNet outperforms both in all 4 tests. This reveals that the key factor contributing to MetaProxNet's success is the more expressive prior relative to PGD.

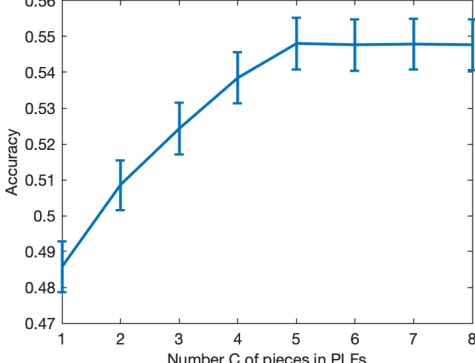

Figure 3: Classification accuracy against the number $C$ of PLF pieces.

## 4.3 IMPACT OF HYPEREPARAMETER $C$

Numerical tests are also carried out to verify the theoretical analysis in Sec. 3.3, which upper bounds the $\ell$-2 error between two PGD optimization outputs: one using the optimal prior and the other using a PLF-induced prior. Specifically, Theorems 3.2 and 3.3 state that this $\ell$-2 error bounds will reduce as $C$ increases, thus offering a better calibrated $\hat{\boldsymbol{\theta}}_t$. To examine the qualities of $\hat{\boldsymbol{\theta}}_t$ with different

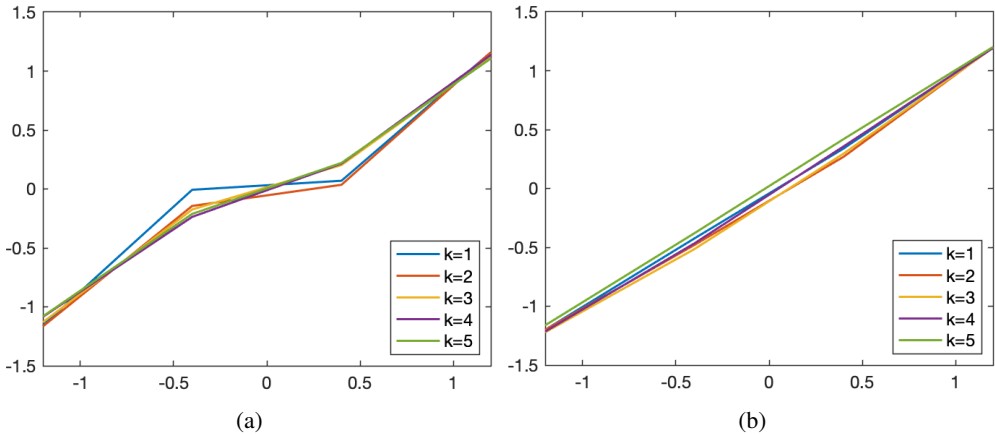

(a)                                                                                              (b)

Figure 4: Visualization of the learned PLFs averaged across CNN layers; (a) first layer; (b) last layer.

$C$, Fig. 3 depicts the test accuracies of MetaProxNet+MAML on 5-class 1-shot miniImageNet as a function of $C$. It can be observed that the accuracy improves with $C$ increasing, which corroborates with our theories. Moreover, $C = 5$ suffices to achieve satisfactory performance, while larger values of $C$ only have a minor impact on MetaProxNet's empirical performance. This suggests that the constants hidden within the error bounds $\mathcal{O}(\frac{1}{C})$ and $\mathcal{O}(\frac{1}{C^2})$ can be small enough in practice. To avoid potential overfitting of priors, we set $C = 5$ in all the tests.

### 4.4 INTERPRETING UNROLLING-BASED PRIORS BY VISUALIZING THE LEARNED PLFS

From an optimization viewpoint, the learned PLFs correspond to an implicit prior pdf that generally comes with no analytical expression. These PLFs can be visualized to further understand the behavior of the unrolled NN. Figs. 4a and 4b respectively depict the averaged $\text{p̌rox}_i^k$ for $i$'s that correspond to the first and last CNN layers. The visualization showcases that the averaged PLF for the first layer is similar to the soft shrinkage function $\mathbb{S}_{\alpha\lambda_i}$ of the sparse prior mentioned in Sec. 3.1, while the last layer tends to have a linear PLF, which resembles that of a Gaussian prior.

In practice, the visualization of the PLFs can be utilized to examine the impact of the prior when updating model parameters, thus guiding the model training process. In Fig. 4, the acquired PLFs keep shallow layer weights being sparse around the initial value $\boldsymbol{\theta}^{\text{init}}$ (that is, less updated) when $k$ is small, while deep layers can be updated freely along its gradient directions. This suggests, when fine-tuning a pre-trained large-scale model on a specific task, it is advisable to freeze the weights of the embedding function and exclusively train the last few layers with a relatively large step size in the initial epochs. Once these deep layers have attained sufficient training, one can then gradually unfreeze the shallow layers and proceed with fine-tuning the entire model. This learned update strategy closely aligns with the widely adopted "gradual unfreezing" training approach for fine-tuning large-scale models, which has been proven effective in various practical applications; see e.g., (Howard & Ruder, 2018).

## 5 CONCLUSIONS AND OUTLOOK

A novel prior information representation approach was pursued in this work using algorithm unrolling to learn more flexible and generalized priors. Under this framework, a meta-learning method termed MetaProxNet was developed with learnable PLFs effecting an implicit prior. The learned prior enjoys interpretability from an optimization vantage point, and can be well explained by visualizing its PLFs. Further, performance analysis established that the PLFs are capable of fitting smooth/continuous proximal functions with a proper selection of $C$. Numerical tests further corroborated empirically the superiority of MetaProxNet relative to meta-learning alternatives in prior representation and learning.

Our future research agenda includes exciting themes on i) investigating various optimizers besides PGD; ii) implementing MetaProxNet with more complicated backbones and DC modules; and, iii) establishing bilevel convergence guarantees for MetaProxNet.

## ACKNOWLEDGMENTS

This work was supported by NSF grants 2102312, 2103256, 2128593, 2126052, and 2212318.

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

## A  PROOF OF THEOREM 3.2.

Smooth $\bar{\mathrm{prox}}_{\mathcal{R}^*,\alpha} \in \mathcal{C}^1([-A, A]^d)$ will be first considered under the following four technical assumptions.

**Assumption A.1.** $\bar{\mathrm{prox}}_{\mathcal{R}^*,\alpha} \in \mathcal{C}^1([-A, A]^d)$ has $G_1$-Lipschitz gradient on $[-A, A]^d$.

**Assumption A.2.** $\bar{\mathcal{L}} \in \mathcal{C}^1([-A, A]^d)$ has $G_2$-Lipschitz gradient on $[-A, A]^d$.

**Assumption A.3.** Constant $A$ is sufficiently large so that $\check{\mathbf{z}}_t^k, \bar{\mathbf{z}}_t^{k*} \in [-A, A]^d$, $\forall t, k$, where $\bar{\mathbf{z}}_t^{k*}$ is the PGD auxiliary variable generated with $\bar{\mathrm{prox}}_{\mathcal{R}^*,\alpha}$.

**Assumption A.4.** Operator $\bar{\mathrm{prox}}_{\mathcal{R}^*,\alpha} \in \mathcal{C}^0([-A, A]^d)$ is $L$-Lipschitz on $[-A, A]^d$.

*Remark* A.5 (Mild assumptions). In Assumption A.1 and A.2, the optimal $\bar{\mathrm{prox}}_{\mathcal{R}^*,\alpha}$ and the loss $\bar{\mathcal{L}}$ are only assumed to be Lipschitz smooth on the compact subset $[-A, A]^d \subset \mathbb{R}^d$, without imposing any strong premise regarding their convexity or Lipschitz continuity.
For Assumption A.3, the existence of such an $A$ can be easily guaranteed when e.g., task-level step size $\alpha \leq 2/G_2$, and level sets $\{\boldsymbol{\theta}_t \mid \bar{\mathcal{L}}(\boldsymbol{\theta}_t; \mathcal{D}_t^{\mathrm{trn}}) \leq \bar{\mathcal{L}}(\mathbf{0}_d; \mathcal{D}_t^{\mathrm{trn}})\}$, $\{\boldsymbol{\theta}_t \mid \check{\mathcal{R}}(\boldsymbol{\theta}_t; \boldsymbol{\theta}) \leq \check{\mathcal{R}}(\mathbf{0}_d; \boldsymbol{\theta})\}$ and $\{\boldsymbol{\theta}_t \mid \mathcal{R}^*(\boldsymbol{\theta}_t) \leq \mathcal{R}^*(\mathbf{0}_d)\}$ are bounded.
In addition, Assumption A.4 can be readily satisfied as well. For example, when $\mathcal{R}^*$ has $G_{\mathcal{R}}$-Lipschitz gradient on $[-A, A]^d$ and $\alpha < 1/G_{\mathcal{R}}$, it follows from the stationary condition of (3) that $\frac{1}{\alpha}(\mathrm{prox}_{\mathcal{R}^*,\alpha}(\mathbf{z}) - \mathbf{z}) + \nabla\mathcal{R}^*(\mathrm{prox}_{\mathcal{R}^*,\alpha}(\mathbf{z})) = 0$. Hence, it holds for $\forall \mathbf{z}, \mathbf{z}' \in [-A, A]^d$ that $\|\mathbf{z} - \mathbf{z}'\|_2 \geq \left|\|\mathrm{prox}_{\mathcal{R}^*,\alpha}(\mathbf{z}) - \mathrm{prox}_{\mathcal{R}^*,\alpha}(\mathbf{z}')\|_2 - \alpha\|\nabla\mathcal{R}^*(\mathrm{prox}_{\mathcal{R}^*,\alpha}(\mathbf{z})) - \nabla\mathcal{R}^*(\mathrm{prox}_{\mathcal{R}^*,\alpha}(\mathbf{z}'))\|_2\right| = (1 - \alpha G_{\mathcal{R}})\|\mathrm{prox}_{\mathcal{R}^*,\alpha}(\mathbf{z}) - \mathrm{prox}_{\mathcal{R}^*,\alpha}(\mathbf{z}')\|_2$. In other words, the Lipschitz constant in this case is upper bounded by $L \leq 1/(1 - \alpha G_{\mathcal{R}})$.

To prove Theorem 3.2, we first show a lemma that is important for bounding the error $|\check{\mathrm{prox}}_i^k - \bar{\mathrm{prox}}_{\mathcal{R}_i^*,\alpha}|$ on $[-A, A]$.

**Lemma A.6.** *Let $f \in \mathcal{C}^1(\mathbb{R}) : \mathbb{R} \mapsto \mathbb{R}$ be a function with $G$-Lipschitz gradient. For $\forall \zeta_1, \zeta_2 \in \mathbb{R}$ and $\zeta_1 \neq \zeta_2$, define*

$$\hat{f}(z) := \frac{(\zeta_2 - z)f(\zeta_1) + (z - \zeta_1)f(\zeta_2)}{\zeta_2 - \zeta_1}. \tag{12}$$

*It then holds for $\forall \gamma \in [0, 1]$ that*

$$\left|f\big((1-\gamma)\zeta_1 + \gamma\zeta_2\big) - \hat{f}\big((1-\gamma)\zeta_1 + \gamma\zeta_2\big)\right| \leq \frac{G}{8}(\zeta_2 - \zeta_1)^2. \tag{13}$$

*Proof.* For notational convenience, let $g(\gamma) := \left|f\big((1-\gamma)\zeta_1 + \gamma\zeta_2\big) - \hat{f}\big((1-\gamma)\zeta_1 + \gamma\zeta_2\big)\right|$. Using the definition of $\hat{f}$ and $g$, it can be easily verified for $\forall \gamma \in (0, 1)$ that $g \in \mathcal{C}^0(\mathbb{R})$ and

$$g(\gamma) \geq g(0) = g(1) = 0. \tag{14}$$

Therefore, there exits at least one maximizer $\gamma^* = \mathrm{argmax}_{\gamma \in (0,1)} g(\gamma)$ inside the open interval $(0, 1)$. For brevity, define the corresponding $\zeta^* = (1 - \gamma^*)\zeta_1 + \gamma^*\zeta_2$. Through Fermat's stationary point theorem (a.k.a. interior extremum theorem), it turns out that $g'(\gamma^*) = 0$, which implies

$$(\zeta_1 - \zeta_2)f'(\zeta^*) = (\zeta_1 - \zeta_2)\hat{f}'(\zeta^*). \tag{15}$$

Since $\zeta_1 \neq \zeta_2$, we obtain

$$f'(\zeta^*) = \hat{f}'(\zeta^*). \tag{16}$$

Next, we discuss the following two possible cases of $\gamma^*$.

**Caes i)** $\gamma^* \in (0, 1/2]$

It follows from (12), (16) and the Lipschitzness of $f'$ that

$$
\begin{aligned}
\left|f'\big((1-\gamma)\zeta_1 + \gamma\zeta_2\big) - \hat{f}'\big((1-\gamma)\zeta_1 + \gamma\zeta_2\big)\right| &= \left|f'\big((1-\gamma)\zeta_1 + \gamma\zeta_2\big) - \hat{f}'(\zeta^*)\right| \\
&= \left|f'\big((1-\gamma)\zeta_1 + \gamma\zeta_2\big) - f'(\zeta^*)\right| \\
&\leq G|(1-\gamma)\zeta_1 + \gamma\zeta_2 - \zeta^*| \\
&= G|\gamma - \gamma^*||\zeta_2 - \zeta_1|. 
\end{aligned}
\tag{17}
$$

As a result, it holds for $\forall \gamma \in [0, 1]$ that

$$
\begin{aligned}
g(\gamma) \leq g(\gamma^*) &\overset{(a)}{=} \left| \int_0^{\gamma^*} \left[ (\zeta_2 - \zeta_1) f'\big((1-\gamma)\zeta_1 + \gamma\zeta_2\big) - (\zeta_2 - \zeta_1)\hat{f}'\big((1-\gamma)\zeta_1 + \gamma\zeta_2\big) \right] d\gamma \right| \\
&\leq |\zeta_2 - \zeta_1| \int_0^{\gamma^*} \left| f'\big((1-\gamma)\zeta_1 + \gamma\zeta_2\big) - \hat{f}'\big((1-\gamma)\zeta_1 + \gamma\zeta_2\big) \right| d\gamma \\
&\overset{(b)}{\leq} G|\zeta_2 - \zeta_1|^2 \int_0^{\gamma^*} (\gamma^* - \gamma) d\gamma \\
&= G(\zeta_2 - \zeta_1)^2 \frac{\gamma^{*2}}{2} \\
&\overset{(c)}{\leq} \frac{G}{8}(\zeta_2 - \zeta_1)^2
\end{aligned}
\tag{18}
$$

where $(a)$ uses the fact that $f(\zeta_1) = \hat{f}(\zeta_1)$, $(b)$ is from (17), and $(c)$ is due to $\gamma^* \leq 1/2$.

**Case ii)** $\quad \gamma^* \in [1/2, 1)$

Likewise, we can also have

$$
\begin{aligned}
\left| f'\big(\eta\zeta_1 + (1-\eta)\zeta_2\big) - \hat{f}'\big(\eta\zeta_1 + (1-\eta)\zeta_2\big) \right| &= \left| f'\big(\eta\zeta_1 + (1-\eta)\zeta_2\big) - \hat{f}'(\zeta^*) \right| \\
&= \left| f'\big(\eta\zeta_1 + (1-\eta)\zeta_2\big) - f'(\zeta^*) \right| \\
&\leq G|\eta\zeta_1 + (1-\eta)\zeta_2 - \zeta^*| \\
&= G|1 - \eta - \gamma^*||\zeta_2 - \zeta_1|.
\end{aligned}
\tag{19}
$$

It then holds for $\forall \gamma \in [0, 1]$ that

$$
\begin{aligned}
g(\gamma) \leq g(\gamma^*) &= \left| \int_{\gamma^*}^1 \left[ (\zeta_2 - \zeta_1) f'\big((1-\gamma)\zeta_1 + \gamma\zeta_2\big) - (\zeta_2 - \zeta_1)\hat{f}'\big((1-\gamma)\zeta_1 + \gamma\zeta_2\big) \right] d\gamma \right| \\
&\leq |\zeta_2 - \zeta_1| \int_{\gamma^*}^1 \left| f'\big((1-\gamma)\zeta_1 + \gamma\zeta_2\big) - \hat{f}'\big((1-\gamma)\zeta_1 + \gamma\zeta_2\big) \right| d\gamma \\
&\overset{(a)}{=} |\zeta_2 - \zeta_1| \int_0^{1-\gamma^*} \left| f'\big(\eta\zeta_1 + (1-\eta)\zeta_2\big) - \hat{f}'\big(\eta\zeta_1 + (1-\eta)\zeta_2\big) \right| d\eta \\
&\overset{(b)}{\leq} G|\zeta_2 - \zeta_1|^2 \int_0^{1-\gamma^*} (1 - \eta - \gamma^*) d\eta \\
&= G(\zeta_2 - \zeta_1)^2 \frac{(1 - \gamma^*)^2}{2} \\
&\leq \frac{G}{8}(\zeta_2 - \zeta_1)^2
\end{aligned}
\tag{20}
$$

where $(a)$ follows by the substitution of integral variable $\gamma = 1 - \eta$, and $(b)$ uses (19).

Combining these two cases with (14) yields the desired conclusion. $\qquad\square$

The next theorem bounds the per-step error $|\,\breve{\text{prox}}_i^k - \bar{\text{prox}}_{\mathcal{R}^*,\alpha}|$ utilizing Lemma A.6.

**Theorem A.7** (Per-step error for smooth proximal operator). *Consider* $\breve{\text{prox}}^k$ *defined by* (6) *with fixed* $\zeta_{i,c}^k = (\frac{2c}{C} - 1)A$. *Define* $\boldsymbol{\psi}_i^k := [\psi_{i,0}^k, \dots, \psi_{i,C}^k]^\top$ *the vector parameterizing* $\breve{\text{prox}}_i^k$. *Then under Assumptions 3.1 and A.1, it holds for* $i = 1, \dots, d$ *and* $k = 1, \dots, K$ *that*

$$
\min_{\boldsymbol{\psi}_i^k} \max_{z \in [-A, A]} \left| \bar{\text{prox}}_{\mathcal{R}_i^*,\alpha}(z) - \breve{\text{prox}}_i^k(z; \boldsymbol{\psi}_i^k) \right| \leq \frac{G_1 A^2}{2C^2}.
\tag{21}
$$

*This bound is tight with the additional constraints that* $\psi_{i,0}^k = \bar{\text{prox}}_{\mathcal{R}_i^*,\alpha}(-A)$ *and* $\psi_{i,C}^k = \bar{\text{prox}}_{\mathcal{R}_i^*,\alpha}(A)$, $\forall k, i$.

*Proof.* Define $\tilde{\psi}_i := [\mathrm{p\bar{r}ox}_{\mathcal{R}_i^*,\alpha}(-A), \mathrm{p\bar{r}ox}_{\mathcal{R}_i^*,\alpha}(-A + \frac{2}{C}A), \ldots, \mathrm{p\bar{r}ox}_{\mathcal{R}_i^*,\alpha}(A)]^\top$ to be the vector collecting the proximal function values at the partition points $\zeta_{i,c}^k = (\frac{2c}{C} - 1)A$. It then follows that

$$\min_{\psi_i^k} \max_{z \in [-A,A]} |\mathrm{p\bar{r}ox}_{\mathcal{R}_i^*,\alpha}(z) - \mathrm{p\check{r}ox}_i^k(z; \psi_i^k)| \leq \max_{z \in [-A,A]} |\mathrm{p\bar{r}ox}_{\mathcal{R}_i^*,\alpha}(z) - \mathrm{p\check{r}ox}_i^k(z; \tilde{\psi}_i)|. \quad (22)$$

Next, applying Lemma A.6 to each piece of $\mathrm{p\check{r}ox}_i^k$, it holds for $\forall \gamma \in [0,1]$ and $c = 1, \ldots, C$ that

$$\left| \mathrm{prox}_{\mathcal{R}_i,\alpha}\left((1-\gamma)\zeta_{i,c-1}^k + \gamma\zeta_{i,c}^k\right) - \mathrm{p\check{r}ox}_i^k\left((1-\gamma)\zeta_{i,c-1}^k + \gamma\zeta_{i,c}^k; \tilde{\psi}_i\right) \right| \leq \frac{G_1}{8}(\zeta_{i,c}^k - \zeta_{i,c-1}^k)^2 = \frac{G_1 A^2}{2C^2}. \quad (23)$$

By noticing that $\cup_{c=1}^C \{(1-\gamma)\zeta_{i,c-1}^k + \gamma\zeta_{i,c}^k \mid \gamma \in [0,1]\} = [\zeta_i^0, \zeta_i^C] = [-A, A]$, we obtain from (23) that

$$|\mathrm{p\bar{r}ox}_{\mathcal{R}_i^*,\alpha}(z) - \mathrm{p\check{r}ox}_i(z; \tilde{\psi}_i)| \leq \frac{G_1 A^2}{2C^2}, \; \forall z \in [-A, A]. \quad (24)$$

Relating (22) to (24) leads to the desired error bound (21).

For later use, we define distance

$$\mathrm{dist}([a,b]; \psi_i) := \max_{z \in [a,b]} \left| \mathrm{p\bar{r}ox}_{\mathcal{R}_i^*,\alpha}(z) - \mathrm{p\check{r}ox}_i^k(z; \psi_i) \right| \quad (25)$$

To illustrate this bound is tight with the additional constraints stated in Theorem A.7, a specific example will be constructed to show that the upper bound can be actually attained. To be more specific, it will be shown that for any given $C \geq 1$, there exists a $\mathrm{p\bar{r}ox}_{\mathcal{R}_i^*,\alpha}$ that satisfies Assumption A.1 and reaches the right side of (21), with the minimizer exactly being

$$\tilde{\psi}_i = \psi_i^* := \operatorname*{argmin}_{\substack{\psi_i: \; \psi_{i,0}=\mathrm{p\bar{r}ox}_{\mathcal{R}_i^*,\alpha}(-A) \\ \psi_{i,C}=\mathrm{p\bar{r}ox}_{\mathcal{R}_i^*,\alpha}(A)}} \mathrm{dist}([-A, A]; \psi_i). \quad (26)$$

For simplicity, we drop the superscript $k$ to write $\zeta_{i,c} = (\frac{2c}{C} - 1)A$ in the sequel. Consider the following proximal function

$$\mathrm{p\bar{r}ox}_{\mathcal{R}_i^*,\alpha}(z) = \begin{cases} 0, & z < -A \\ \frac{G_1}{2}(z - \zeta_{i,c})^2 + \frac{2G_1 A^2}{C^2}c, & \zeta_{i,c} \leq z < \zeta_{i,c+1}, \; c = 0, 2, \ldots, 2\lfloor \frac{C}{2} \rfloor \\ -\frac{G_1}{2}(z - \zeta_{i,c+1})^2 + \frac{2G_1 A^2}{C^2}(c+1), & \zeta_{i,c} \leq z < \zeta_{i,c+1}, \; c = 1, 3, \ldots, 2\lfloor \frac{C+1}{2} \rfloor - 1 \\ 2G_1 A(1 - \frac{2}{C}\lfloor \frac{C}{2} \rfloor)(z - A) + \frac{2G_1 A^2}{C^2}, & z \geq A \end{cases} \quad (27)$$

It can be verified that this function satisfies Assumption A.1 by showing that

$$\mathrm{p\bar{r}ox}'_{\mathcal{R}_i^*,\alpha}(z) = \begin{cases} 0, & z < -A \\ G_1(z - \zeta_{i,c}), & \zeta_{i,c} \leq z < \zeta_{i,c+1}, \; c = 0, 2, \ldots, 2\lfloor \frac{C}{2} \rfloor \\ G_1(\zeta_{i,c+1} - z), & \zeta_{i,c} \leq z < \zeta_{i,c+1}, \; c = 1, 3, \ldots, 2\lfloor \frac{C+1}{2} \rfloor - 1 \\ 2G_1 A(1 - \frac{2}{C}\lfloor \frac{C}{2} \rfloor), & z \geq A \end{cases} \quad (28)$$

is continuous, and the second-order derivate

$$|\mathrm{p\bar{r}ox}''_{\mathcal{R}_i^*,\alpha}(z)| = \begin{cases} G_1, & -A \leq z < A \\ 0, & \text{otherwise} \end{cases} \quad (29)$$

is bounded by $G_1$.

In such case, the $c$-th element of $\tilde{\psi}_i$ is $\tilde{\psi}_{i,c} = \mathrm{p\bar{r}ox}_{\mathcal{R}_i^*,\alpha}(\zeta_{i,c}) = \frac{2G_1 A^2}{C^2}c$. It then follows from (6) that

$$\mathrm{p\check{r}ox}_i^k(z; \tilde{\psi}_i) = \frac{G_1 A}{C}(z + A). \quad (30)$$

As a result, one can have

$$\mathrm{dist}([-A, A]; \tilde{\psi}_i) = \mathrm{dist}([\zeta_{i,c-1}, \zeta_{i,c}]; \tilde{\psi}_i) = \frac{G_1 A^2}{2C^2}, \quad (31)$$

where the maximum (hidden inside $\mathrm{dist}$; cf. (25)) is attained at $z = \frac{\zeta_{i,c-1}+\zeta_{i,c}}{2}$ for each $c = 1, \ldots, C$.

What remains now is to prove (26), which relies on the mathematical induction of $C$. The proof starts with the base case that $C = 1$. With the two extra constraints in effect, we already have

$$\boldsymbol{\psi}_i^* = \underset{\substack{\boldsymbol{\psi}_i:\ \psi_{i,0}=\text{pr\={o}x}_{\mathcal{R}_i^*,\alpha}(-A) \\ \psi_{i,C}=\text{pr\={o}x}_{\mathcal{R}_i^*,\alpha}(A)}}{\text{argmin}} \text{dist}([-A,A];\boldsymbol{\psi}_i) = [\text{pr\={o}x}_{\mathcal{R}_i^*,\alpha}(-A), \text{pr\={o}x}_{\mathcal{R}_i^*,\alpha}(A)]^\top = \tilde{\boldsymbol{\psi}}_i \quad (32)$$

and the minimum value $\text{dist}([-A,A];\boldsymbol{\psi}_i^*) = \frac{G_1 A^2}{2C^2}$.

Now, with the inductive hypothesis that (26) holds for $C = 1,\ldots,C'$ ($C' \geq 1$), we now prove that (26) is also true for $C = C' + 1$. Without loss of generality, assume $C'$ is even so that $C' + 1$ is odd. A similar analysis can be readily carried out when $C'$ is odd.

Next, we discuss the following three possible cases to determine the optimal $\boldsymbol{\psi}_i^*$ that minimizes $\text{dist}([-A,A];\boldsymbol{\psi}_i)$. In particular, it will be proved that the minimum distance of $\frac{G_1 A^2}{2C^2}$ can be reached only in the first case, where the induction hypothesis for $C = C' + 1$ holds.

**Case i)** $[\boldsymbol{\psi}_i^*]_{C'} = [\tilde{\boldsymbol{\psi}}_i]_{C'} = \text{pr\={o}x}_{\mathcal{R}_i^*,\alpha}(\zeta_{i,C'})$. With this additional condition, it holds that

$$\boldsymbol{\psi}_i^* = \underset{\substack{\boldsymbol{\psi}_i:\ \psi_{i,0}=\text{pr\={o}x}_{\mathcal{R}_i^*,\alpha}(-A) \\ \psi_{i,C}=\text{pr\={o}x}_{\mathcal{R}_i^*,\alpha}(A) \\ \psi_{i,C'}=\text{pr\={o}x}_{\mathcal{R}_i^*,\alpha}(\zeta_{i,C'})}}{\text{argmin}} \text{dist}([-A,A];\boldsymbol{\psi}_i)$$

$$= \underset{\substack{\boldsymbol{\psi}_i:\ \psi_{i,0}=\text{pr\={o}x}_{\mathcal{R}_i^*,\alpha}(-A) \\ \psi_{i,C}=\text{pr\={o}x}_{\mathcal{R}_i^*,\alpha}(A) \\ \psi_{i,C'}=\text{pr\={o}x}_{\mathcal{R}_i^*,\alpha}(\zeta_{i,C'})}}{\text{argmin}} \max\left\{ \text{dist}([-A,\zeta_{i,C'}];\boldsymbol{\psi}_i),\ \text{dist}([\zeta_{i,C'},A];\boldsymbol{\psi}_i) \right\} \quad (33)$$

where the last equality follows from the definition (25).

By applying the base inductive case (i.e., $C = 1$) on the interval $[\zeta_{i,C'}, A]$ that contains one piece of $\text{pr\v{o}x}_i^k$ (note that $\zeta_{i,C'+1} = \zeta_{i,C} = A$ by definition), it follows that

$$\underset{\substack{\boldsymbol{\psi}_i:\ \psi_{i,C'}=\text{pr\={o}x}_{\mathcal{R}_i^*,\alpha}(\zeta_{i,C'}) \\ \psi_{i,C}=\text{pr\={o}x}_{\mathcal{R}_i^*,\alpha}(\zeta_{i,C})}}{\min} \text{dist}([\zeta_{i,C'},A];\boldsymbol{\psi}_i) = \frac{G_1(A-\zeta_{i,C'})^2}{8} = \frac{G_1 A^2}{2C^2}, \quad (34)$$

where the second equality utilizes that $\{\zeta_{i,c}\}_{c=0}^C$ uniformly partition $[-A,A]$ so that $A - \zeta_{i,C'} = \zeta_{i,C'+1} - \zeta_{i,C'} = \frac{2A}{C}$.

Moreover, using the inductive hypothesis for $C = C'$ on the interval $[-A,\zeta_{i,C'}]$ containing $C'$ pieces of $\text{pr\v{o}x}_i^k$, we obtain

$$\underset{\substack{\boldsymbol{\psi}_i:\ \psi_{i,0}=\text{pr\={o}x}_{\mathcal{R}_i^*,\alpha}(-A) \\ \psi_{i,C'}=\text{pr\={o}x}_{\mathcal{R}_i^*,\alpha}(\zeta_{i,C'})}}{\min} \text{dist}([-A,\zeta_{i,C'}];\boldsymbol{\psi}_i) = \frac{G_1(\zeta_{i,C'}+A)^2}{8C'^2} = \frac{G_1 A^2}{2C^2} \quad (35)$$

and that the first $C'$ elements of the minimizer of (35) are equal to $[\tilde{\boldsymbol{\psi}}_i]_{1:C'}$.

Therefore, combining (33)-(35) we arrive at

$$\boldsymbol{\psi}_i^* = [[\tilde{\boldsymbol{\psi}}_i]_{1:C'}^\top, \text{pr\={o}x}_{\mathcal{R}_i^*,\alpha}(A)]^\top = \tilde{\boldsymbol{\psi}}_i, \quad (36)$$

which indicates that the inductive hypothesis (26) also holds for $C = C' + 1$.

Next, we show that the minimum distance of (26) in the following two cases is larger than $\frac{G_1 A^2}{2C^2}$.

**Case ii)** $\psi_{i,C'}^* > \tilde{\psi}_{i,C'} = \text{pr\={o}x}_{\mathcal{R}_i^*,\alpha}(\zeta_{i,C'})$. According to (27), (30) and that $C'$ is even, one can easily verify that

$$\text{pr\={o}x}_{\mathcal{R}_i^*,\alpha}(z) - \text{pr\v{o}x}_i^k(z;\tilde{\boldsymbol{\psi}}_i) \geq 0,\ z \in [\zeta_{i,C'-1}, \zeta_{i,C'}] \quad (37a)$$

$$\text{pr\={o}x}_{\mathcal{R}_i^*,\alpha}(z) - \text{pr\v{o}x}_i^k(z;\tilde{\boldsymbol{\psi}}_i) \leq 0,\ z \in [\zeta_{i,C'}, A]. \quad (37b)$$

Since $\psi_{i,C'}^* > \tilde{\psi}_{i,C'}$ and $\psi_{i,C}^* = \tilde{\psi}_{i,C}$, it follows from the definition (6) that

$$\text{p}\check{\text{r}}\text{ox}_i^k(z; \boldsymbol{\psi}_i^*) > \text{p}\check{\text{r}}\text{ox}_i^k(z; \tilde{\boldsymbol{\psi}}_i), \ z \in [\zeta_{i,C'}, A]. \tag{38}$$

Then, it holds that

$$\begin{aligned}
\frac{G_1 A^2}{2C^2} &= \text{dist}([\zeta_{i,C'}, A]; \tilde{\boldsymbol{\psi}}_i) = \max_{z \in [\zeta_{i,C'}, A]} \left| \text{p}\bar{\text{r}}\text{ox}_{\mathcal{R}_i^*, \alpha}(z) - \text{p}\check{\text{r}}\text{ox}_i^k(z; \tilde{\boldsymbol{\psi}}_i) \right| \\
&\overset{(a)}{=} \left| \text{p}\bar{\text{r}}\text{ox}_{\mathcal{R}_i^*, \alpha}\left(\frac{\zeta_{i,C'} + A}{2}\right) - \text{p}\check{\text{r}}\text{ox}_i^k\left(\frac{\zeta_{i,C'} + A}{2}; \tilde{\boldsymbol{\psi}}_i\right) \right| \\
&\overset{(b)}{=} \text{p}\check{\text{r}}\text{ox}_i^k\left(\frac{\zeta_{i,C'} + A}{2}; \tilde{\boldsymbol{\psi}}_i\right) - \text{p}\bar{\text{r}}\text{ox}_{\mathcal{R}_i^*, \alpha}\left(\frac{\zeta_{i,C'} + A}{2}\right) \\
&\overset{(c)}{<} \text{p}\check{\text{r}}\text{ox}_i^k\left(\frac{\zeta_{i,C'} + A}{2}; \boldsymbol{\psi}_i^*\right) - \text{p}\bar{\text{r}}\text{ox}_{\mathcal{R}_i^*, \alpha}\left(\frac{\zeta_{i,C'} + A}{2}\right) \\
&\leq \max_{z \in [\zeta_{i,C'}, A]} \left| \text{p}\bar{\text{r}}\text{ox}_{\mathcal{R}_i^*, \alpha}(z) - \text{p}\check{\text{r}}\text{ox}_i^k(z; \boldsymbol{\psi}_i^*) \right| = \text{dist}([\zeta_{i,C'}, A]; \boldsymbol{\psi}_i^*),
\end{aligned} \tag{39}$$

where $(a)$ uses that (31) is achieved at $z = \frac{\zeta_{i,c-1} + \zeta_{i,c}}{2}$, $(b)$ follows from (37b), and $(c)$ is due to (38). In other words, if $\psi_{i,C'}^* > \tilde{\psi}_{i,C'}$, it must hold that

$$\text{dist}([\zeta_i^0, \zeta_i^C]; \boldsymbol{\psi}_i^*) > \frac{G_1(\zeta_i^C - \zeta_i^0)^2}{8C^2}$$

, which is larger than that of case i). Therefore, the optimal $\boldsymbol{\psi}_i^*$ must satisfy $\psi_{i,C'}^* \leq \tilde{\psi}_{i,C'}$.

**Case iii)** $\quad \psi_{i,C'}^* < \tilde{\psi}_{i,C'} = \text{p}\bar{\text{r}}\text{ox}_{\mathcal{R}_i^*, \alpha}(\zeta_{i,C'})$. Again with (31), one can easily get

$$\begin{aligned}
\frac{G_1 A^2}{2C^2} &= \text{dist}([\zeta_{i,C'-1}, \zeta_{i,C'}]; \tilde{\boldsymbol{\psi}}_i) \\
&= \text{p}\bar{\text{r}}\text{ox}_{\mathcal{R}_i^*, \alpha}\left(\frac{\zeta_{i,C'-1} + \zeta_{i,C'}}{2}\right) - \text{p}\check{\text{r}}\text{ox}_i^k\left(\frac{\zeta_{i,C'-1} + \zeta_{i,C'}}{2}; \tilde{\boldsymbol{\psi}}_i\right).
\end{aligned} \tag{40}$$

Recall from the definition (6) that $\text{p}\check{\text{r}}\text{ox}_i^k(z; \boldsymbol{\psi}_i)$, $z \in [\zeta_{i,C'-1}, \zeta_{i,C'}]$ is defined as the line segment connecting points $(\zeta_{i,C'-1}, \psi_{i,C'-1})$ and $(\zeta_{i,C'}, \psi_{i,C'})$. To ensure $\text{dist}([\zeta_{i,C'-1}, \zeta_{i,C'}]; \boldsymbol{\psi}_i^*) \leq \frac{G_1 A^2}{2C^2}$, a necessary condition is $\text{p}\check{\text{r}}\text{ox}_i^k\left(\frac{\zeta_{i,C'-1} + \zeta_{i,C'}}{2}; \boldsymbol{\psi}_i^*\right) \geq \text{p}\check{\text{r}}\text{ox}_i^k\left(\frac{\zeta_{i,C'-1} + \zeta_{i,C'}}{2}; \tilde{\boldsymbol{\psi}}_i\right)$; cf. (37a). Since we have $\psi_{i,C'}^* < \tilde{\psi}_{i,C'}$ in this case, it must hold that $\psi_{i,C'-1}^* > \tilde{\psi}_{i,C'-1}$. By applying this analysis recursively, one can proceed to obtain a series of necessary conditions of $\text{dist}([-A, A]; \boldsymbol{\psi}_i^*) \leq \frac{G_1 A^2}{2C^2}$, which are (recall that $C'$ is presumed even)

$$\psi_{i,C'-1}^* > \tilde{\psi}_{i,C'-1}, \ \ \psi_{i,C'-2}^* < \tilde{\psi}_{i,C'-2}, \tag{41}$$

$$\cdots, \tag{42}$$

$$\psi_{i,1}^* > \tilde{\psi}_{i,1}, \qquad \psi_{i,0}^* < \tilde{\psi}_{i,0}. \tag{43}$$

This contradicts with the constraint that $\psi_{i,0}^* = \text{p}\bar{\text{r}}\text{ox}_{\mathcal{R}_i^*, \alpha}(-A) = \tilde{\psi}_{i,0}$; cf. (26). That is to say, requiring $\psi_{i,C'}^* < \tilde{\psi}_{i,C'}$ will lead to $\text{dist}([-A, A]; \boldsymbol{\psi}_i^*) > \frac{G_1 A^2}{2C^2}$, which is not optimal.

To the end, through the three cases we conclude that the minimizer $\boldsymbol{\psi}_i^*$ must satisfy $\psi_{i,C'}^* = \tilde{\psi}_{i,C'}$, which implies (26) holds for $C = C' + 1$; see (36). The proof is thus completed. $\qquad \square$

Building upon the per-step error bound established in Theorem A.7, the $K$-step cumulative error bound will next be proved. In particular, the following theorem offers an upper bound for the normalized error on (shifted) $\hat{\boldsymbol{\theta}}_t$.

**Theorem A.8** (Formal statement: finite-step PGD error for smooth proximal operators). *Consider* $\text{p}\check{\text{r}}\text{ox}^k$ *defined by* (6) *with fixed* $\zeta_{i,c}^k = (\frac{2c}{C} - 1)A$. *Define* $\boldsymbol{\Psi} := [\boldsymbol{\psi}^1, \ldots, \boldsymbol{\psi}^K]$ *the matrix parameterizing* $\{\text{p}\check{\text{r}}\text{ox}^k\}_{k=1}^K$. *Let* $\bar{\boldsymbol{\theta}}_t^K$ *and* $\check{\boldsymbol{\theta}}_t^K$ *be the K-step PGD outputs using* $\text{p}\bar{\text{r}}\text{ox}_{\mathcal{R}^*, \alpha}$ *and* $\text{p}\check{\text{r}}\text{ox}^k$,

*respectively. With Assumptions 3.1 and A.1-A.4 in effect, it holds for $t = 1, \ldots, T$ that*

$$\min_{\Psi} \frac{1}{\sqrt{d}} \big\| \bar{\boldsymbol{\theta}}_t^K - \check{\boldsymbol{\theta}}_t^K(\Psi) \big\|_2 = \mathcal{O}(\frac{1}{C^2}). \tag{44}$$

*This bound is tight with the additional constraints that $\psi_{i,0}^k = \mathrm{pr\bar{o}x}_{\mathcal{R}_i^*, \alpha}(-A)$ and $\psi_{i,C}^k = \mathrm{pr\bar{o}x}_{\mathcal{R}_i^*, \alpha}(A), \ \forall k, i.$*

*Proof.* For notational compactness, define $\tilde{\boldsymbol{\psi}}^k := \mathrm{argmin}_{\boldsymbol{\psi}^k} \max_{-A\mathbf{1}_d \preceq \mathbf{z} \preceq A\mathbf{1}_d} \| \mathrm{pr\bar{o}x}_{\mathcal{R}^*, \alpha}(\mathbf{z}) - \mathrm{pr\check{o}x}^k(\mathbf{z}; \boldsymbol{\psi}^k) \|_2$ and $\tilde{\boldsymbol{\Psi}} := [\tilde{\boldsymbol{\psi}}^1, \ldots, \tilde{\boldsymbol{\psi}}^K]$. Since both $\mathrm{pr\bar{o}x}_{\mathcal{R}^*, \alpha}$ and $\mathrm{pr\check{o}x}^k$ are factorable across the dimensions of their inputs, we know that $\tilde{\boldsymbol{\psi}}^k$ is the concatenation of the minimizers

$$\tilde{\psi}_i^k = \mathrm{argmin}_{\boldsymbol{\psi}_i^k} \max_{z \in [-A, A]} \| \mathrm{pr\bar{o}x}_{\mathcal{R}_i^*, \alpha}(z) - \mathrm{pr\check{o}x}_i^k(z; \psi_i^k) \|_2, \ i = 1, \ldots, d$$

for each dimension. It then holds for $k = 1, \ldots, K$ that

$$\min_{\Psi} \big\| \bar{\boldsymbol{\theta}}_t^k - \check{\boldsymbol{\theta}}_t^k(\Psi) \big\|_2$$

$$\leq \big\| \bar{\boldsymbol{\theta}}_t^k - \check{\boldsymbol{\theta}}_t^k(\tilde{\boldsymbol{\Psi}}) \big\|_2 = \big\| \mathrm{pr\bar{o}x}_{\mathcal{R}^*, \alpha}(\bar{\mathbf{z}}_t^k) - \mathrm{pr\check{o}x}_i^k(\check{\mathbf{z}}_t^k; \tilde{\boldsymbol{\psi}}^k) \big\|_2$$

$$\leq \big\| \mathrm{pr\bar{o}x}_{\mathcal{R}^*, \alpha}(\bar{\mathbf{z}}_t^k) - \mathrm{pr\bar{o}x}_{\mathcal{R}^*, \alpha}(\check{\mathbf{z}}_t^k) \big\|_2 + \big\| \mathrm{pr\bar{o}x}_{\mathcal{R}^*, \alpha}(\check{\mathbf{z}}_t^k) - \mathrm{pr\check{o}x}_i^k(\check{\mathbf{z}}_t^k; \tilde{\boldsymbol{\psi}}^k) \big\|_2$$

$$\overset{(a)}{\leq} \big\| \mathrm{pr\bar{o}x}_{\mathcal{R}^*, \alpha}(\bar{\mathbf{z}}_t^k) - \mathrm{pr\bar{o}x}_{\mathcal{R}^*, \alpha}(\check{\mathbf{z}}_t^k) \big\|_2 + \frac{\sqrt{d} A^2 G_1}{2C^2}$$

$$\overset{(b)}{\leq} L \big\| \bar{\mathbf{z}}_t^k - \check{\mathbf{z}}_t^k(\tilde{\boldsymbol{\Psi}}) \big\|_2 + \frac{\sqrt{d} A^2 G_1}{2C^2}$$

$$\overset{(c)}{\leq} L \big\| \bar{\boldsymbol{\theta}}_t^{k-1} - \check{\boldsymbol{\theta}}_t^{k-1}(\tilde{\boldsymbol{\Psi}}) \big\|_2 + \alpha L \big\| \nabla_{\bar{\boldsymbol{\theta}}_t^{k-1}} \bar{\mathcal{L}}(\bar{\boldsymbol{\theta}}_t^{k-1}; \mathcal{D}_t^{\mathrm{trn}}) - \nabla_{\check{\boldsymbol{\theta}}_t^{k-1}} \bar{\mathcal{L}}(\check{\boldsymbol{\theta}}_t^{k-1}; \mathcal{D}_t^{\mathrm{trn}}) \big\|_2 + \frac{\sqrt{d} A^2 G_1}{2C^2}$$

$$\overset{(d)}{\leq} L(1 + \alpha G_2) \big\| \bar{\boldsymbol{\theta}}_t^{k-1} - \check{\boldsymbol{\theta}}_t^{k-1}(\tilde{\boldsymbol{\Psi}}) \big\|_2 + \frac{\sqrt{d} A^2 G_1}{2C^2} \tag{45}$$

where $(a)$ follows from Theorem A.7 and Assumptions A.1-A.3, $(b)$ uses Assumption A.4, $(c)$ is from (4) and (7), and $(d)$ is due to Assumption A.2.

Using this recursive relationship between $\| \bar{\boldsymbol{\theta}}_t^k - \check{\boldsymbol{\theta}}_t^k(\tilde{\boldsymbol{\Psi}}) \|_2$ (line 2) and $\| \bar{\boldsymbol{\theta}}_t^{k-1} - \check{\boldsymbol{\theta}}_t^{k-1}(\tilde{\boldsymbol{\Psi}}) \|_2$ (the last line), together with the boundary condition $\| \bar{\boldsymbol{\theta}}_t^0 - \check{\boldsymbol{\theta}}_t^0 \|_2 = \| \mathbf{0}_d - \mathbf{0}_d \|_2 = 0$, we arrive at the solution

$$\big\| \bar{\boldsymbol{\theta}}_t^k - \check{\boldsymbol{\theta}}_t^k(\tilde{\boldsymbol{\Psi}}) \big\|_2 \leq \begin{cases} \frac{1 - L^k(1 + \alpha G_2)^k}{1 - L(1 + \alpha G_2)} \frac{\sqrt{d} A^2 G_1}{2C^2}, & \text{if } L(1 + \alpha G_2) \neq 1 \\ k \frac{\sqrt{d} A^2 G_1}{2C^2}, & \text{otherwise} \end{cases} = \mathcal{O}(\frac{\sqrt{d}}{C^2}). \tag{46}$$

Dividing by $\sqrt{d}$ and minimizing over $\Psi$ on both side of $\| \bar{\boldsymbol{\theta}}_t^k - \check{\boldsymbol{\theta}}_t^k(\Psi) \|_2 \leq \| \bar{\boldsymbol{\theta}}_t^k - \check{\boldsymbol{\theta}}_t^k(\tilde{\boldsymbol{\Psi}}) \|_2$ lead to

$$\min_{\Psi} \frac{1}{\sqrt{d}} \big\| \bar{\boldsymbol{\theta}}_t^k - \check{\boldsymbol{\theta}}_t^k(\Psi) \big\|_2 \leq \frac{1}{\sqrt{d}} \big\| \bar{\boldsymbol{\theta}}_t^k - \check{\boldsymbol{\theta}}_t^k(\tilde{\boldsymbol{\Psi}}) \big\|_2 = \mathcal{O}(\frac{1}{C^2}). \tag{47}$$

Plugging in (46) with $k = K$ gives (9).

In addition, the tightness of (9) follows from the tightness of Theorem A.7. $\square$

# B PROOF OF THEOREM 3.3

The proof of Theorem 3.3 relies on a more generic lemma which can be applied to non-smooth (but still Lipschitz) $\mathrm{pr\bar{o}x}_{\mathcal{R}^*, \alpha}$.

**Lemma B.1.** *Let $f \in \mathcal{C}^0(\mathbb{R}) : \mathbb{R} \mapsto \mathbb{R}$ be an $L$-Lipschitz function. For $\forall \zeta_1, \zeta_2 \in \mathbb{R}$ and $\zeta_1 \neq \zeta_2$, define*

$$\hat{f}(z) := \frac{(\zeta_2 - z) f(\zeta_1) + (z - \zeta_1) f(\zeta_2)}{\zeta_2 - \zeta_1}. \tag{48}$$

*It then holds for $\forall \gamma \in [0,1]$ that*

$$\left| f\big((1-\gamma)\zeta_1 + \gamma\zeta_2\big) - \hat{f}\big((1-\gamma)\zeta_1 + \gamma\zeta_2\big) \right| \leq \frac{L}{2}|\zeta_2 - \zeta_1|. \tag{49}$$

*Proof.* Define $g(\gamma) := \left| f\big((1-\gamma)\zeta_1 + \gamma\zeta_2\big) - \hat{f}\big((1-\gamma)\zeta_1 + \gamma\zeta_2\big) \right|$. Following the same step (14) of Lemma A.6, it can be shown that there also exists at least one maximizer $\gamma^* \in (0,1)$ of $g(\gamma)$.

Thus we obtain

$$
\begin{aligned}
g(\gamma) \leq g(\gamma^*) &= \left| f\big((1-\gamma^*)\zeta_1 + \gamma^*\zeta_2\big) - \hat{f}\big((1-\gamma^*)\zeta_1 + \gamma^*\zeta_2\big) \right| \\
&\overset{(a)}{=} \left| f\big((1-\gamma^*)\zeta_1 + \gamma^*\zeta_2\big) - (1-\gamma^*)f(\zeta_1) - \gamma^* f(\zeta_2) \right| \\
&\leq (1-\gamma^*)\left| f\big((1-\gamma^*)\zeta_1 + \gamma^*\zeta_2\big) - f(\zeta_1) \right| + \gamma^*\left| f\big((1-\gamma^*)\zeta_1 + \gamma^*\zeta_2\big) - f(\zeta_2) \right| \\
&\overset{(b)}{\leq} 2\gamma^*(1-\gamma^*)L|\zeta_2 - \zeta_1| \\
&\overset{(c)}{\leq} \frac{L}{2}|\zeta_2 - \zeta_1|,
\end{aligned}
\tag{50}
$$

where $(a)$ follows from the definition (48) of $\hat{f}$, $(b)$ exploits the Lipschitzness of $f$, and $(c)$ is due to that $\gamma^*(1-\gamma^*) \leq 1/4$ for $\gamma^* \in (0,1)$. $\qquad\square$

With Lemma B.1 at hand, Theorem 3.3 can be proved using similar techniques as Theorem 3.2.

**Theorem B.2** (Formal statement: finite-step PGD error for continuous proximal operators). *Consider the notations defined in Theorem 3.2. With Assumptions 3.1 and A.2-A.4 in effect, it holds for $t = 1, \ldots, T$ that*

$$\min_{\boldsymbol{\Psi}} \frac{1}{\sqrt{d}} \left\| \bar{\boldsymbol{\theta}}_t^K - \check{\boldsymbol{\theta}}_t^K(\boldsymbol{\Psi}) \right\|_2 = \mathcal{O}(\frac{1}{C}). \tag{51}$$

*This bound is tight with the additional constraints that $\psi_{i,0}^k = \mathrm{prox}_{\mathcal{R}_i^*,\alpha}(-A)$ and $\psi_{i,C}^k = \mathrm{prox}_{\mathcal{R}_i^*,\alpha}(A)$, $\forall k, i$.*

*Proof.* Following the same steps of Theorem A.7, it can be shown that the per-step error bound for continuous $\mathrm{prox}_{\mathcal{R}^*,\alpha}$ is

$$\min_{\boldsymbol{\psi}_i^k} \max_{z \in [-A,A]} \left| \mathrm{prox}_{\mathcal{R}_i^*,\alpha}(z) - \check{\mathrm{prox}}_i^k(z; \boldsymbol{\psi}_i^k) \right| \leq \frac{AL}{C}, \ \forall i. \tag{52}$$

Also, this bound is tight provided with the additional constraints that $\psi_{i,0}^k = \mathrm{prox}_{\mathcal{R}_i^*,\alpha}(-A)$ and $\psi_{i,C} = \mathrm{prox}_{\mathcal{R}_i^*,\alpha}(A)$.

Likewise, we define $\tilde{\boldsymbol{\psi}}^k := \mathrm{argmin}_{\boldsymbol{\psi}^k} \max_{-A\mathbf{1}_d \preceq \mathbf{z} \preceq A\mathbf{1}_d} \left\| \mathrm{prox}_{\mathcal{R}^*,\alpha}(\mathbf{z}) - \check{\mathrm{prox}}^k(\mathbf{z}; \boldsymbol{\psi}^k) \right\|_2$ and $\tilde{\boldsymbol{\Psi}} := [\tilde{\boldsymbol{\psi}}^1, \ldots, \tilde{\boldsymbol{\psi}}^K]$. Then, it follows from the first two lines of (45) that

$$
\begin{aligned}
\min_{\boldsymbol{\Psi}} \left\| \bar{\boldsymbol{\theta}}_t^k - \check{\boldsymbol{\theta}}_t^k(\boldsymbol{\Psi}) \right\|_2 &\leq \left\| \mathrm{prox}_{\mathcal{R}^*,\alpha}(\bar{\mathbf{z}}_t^k) - \mathrm{prox}_{\mathcal{R}^*,\alpha}(\check{\mathbf{z}}_t^k) \right\|_2 + \left\| \mathrm{prox}_{\mathcal{R}^*,\alpha}(\check{\mathbf{z}}_t^k) - \check{\mathrm{prox}}^k(\check{\mathbf{z}}_t^k; \tilde{\boldsymbol{\psi}}^k) \right\|_2 \\
&\overset{(a)}{\leq} L\left\| \bar{\mathbf{z}}_t^k - \check{\mathbf{z}}_t^k(\tilde{\boldsymbol{\Psi}}) \right\|_2 + \frac{\sqrt{d}AL}{C} \\
&\overset{(b)}{\leq} L(1+\alpha G_2)\left\| \bar{\boldsymbol{\theta}}_t^{k-1} - \check{\boldsymbol{\theta}}_t^{k-1}(\tilde{\boldsymbol{\Psi}}) \right\|_2 + \frac{\sqrt{d}AL}{C},
\end{aligned}
\tag{53}
$$

where $(a)$ is from Assumption A.4 and (52), and $(b)$ utilizes (4), (7) and Assumption A.2.

To the end, this recursive relationship, combined with the condition $\|\bar{\boldsymbol{\theta}}_t^0 - \check{\boldsymbol{\theta}}_t^0\|_2 = 0$, results in

$$\min_{\boldsymbol{\Psi}} \left\| \bar{\boldsymbol{\theta}}_t^K - \check{\boldsymbol{\theta}}_t^K(\boldsymbol{\Psi}) \right\|_2 \leq \begin{cases} \frac{1-L^K(1+\alpha G_2)^K}{1-L(1+\alpha G_2)} \frac{\sqrt{d}AL}{C}, & \text{if } L(1+\alpha G_2) \neq 1 \\ K\frac{\sqrt{d}AL}{C}, & \text{otherwise} \end{cases} = \mathcal{O}(\frac{\sqrt{d}}{C}). \tag{54}$$

Dividing both sides by $\sqrt{d}$ completes the proof. $\qquad\square$

## C   Upper bound of (10)

As discussed in Sec. 3.3, the optimal $\boldsymbol{\Psi}^*$ that minimizes (9) and (11) is typically unavailable. An feasible approximation is the sub-optimal $\hat{\boldsymbol{\Psi}}$ obtained from the ERM (1), which brings about an extra error term $\frac{1}{\sqrt{d}}\big\|\breve{\boldsymbol{\theta}}_t^K(\hat{\boldsymbol{\Psi}}) - \breve{\boldsymbol{\theta}}_t^K(\boldsymbol{\Psi}^*)\big\|_2$; cf. (10). This section derives its upper bound, based on the following extra assumption.

**Assumption C.1.** $\breve{\mathrm{prox}}^k(\mathbf{z}; \boldsymbol{\psi}^k) \in \mathcal{C}^0([-A, A]^d)$ is $L_1$- and $L_2$-Lipschitz w.r.t. $\mathbf{z}$ and $\boldsymbol{\psi}^k$ for $-A\mathbf{1}_d \preceq \mathbf{z} \preceq A\mathbf{1}_d$, respectively.

Denoting by $\hat{\boldsymbol{\psi}}^k$ and $\boldsymbol{\psi}^{*k}$ the $k$-th columns of $\hat{\boldsymbol{\Psi}}$ and $\boldsymbol{\Psi}^*$ that parametrize $\breve{\mathrm{prox}}^k$, it holds for $k = 1, \ldots, K$ that

$$\big\|\breve{\boldsymbol{\theta}}_t^k(\hat{\boldsymbol{\Psi}}) - \breve{\boldsymbol{\theta}}_t^k(\boldsymbol{\Psi}^*)\big\|_2$$

$$\overset{(a)}{=} \|\breve{\mathrm{prox}}^k(\breve{\mathbf{z}}_t^k(\hat{\boldsymbol{\Psi}}); \hat{\boldsymbol{\psi}}^k) - \breve{\mathrm{prox}}^k(\breve{\mathbf{z}}_t^k(\boldsymbol{\Psi}^*); \boldsymbol{\psi}^{*k})\|_2$$

$$\leq \|\breve{\mathrm{prox}}^k(\breve{\mathbf{z}}_t^k(\hat{\boldsymbol{\Psi}}); \hat{\boldsymbol{\psi}}^k) - \breve{\mathrm{prox}}^k(\breve{\mathbf{z}}_t^k(\hat{\boldsymbol{\Psi}}); \boldsymbol{\psi}^{*k})\|_2 + \|\breve{\mathrm{prox}}^k(\breve{\mathbf{z}}_t^k(\hat{\boldsymbol{\Psi}}); \boldsymbol{\psi}^{*k}) - \breve{\mathrm{prox}}^k(\breve{\mathbf{z}}_t^k(\boldsymbol{\Psi}^*); \boldsymbol{\psi}^{*k})\|_2$$

$$\overset{(b)}{\leq} L_2\|\hat{\boldsymbol{\psi}}^k - \boldsymbol{\psi}^{*k}\|_2 + L_1\|\breve{\mathbf{z}}_t^k(\hat{\boldsymbol{\Psi}}) - \breve{\mathbf{z}}_t^k(\boldsymbol{\Psi}^*)\|_2$$

$$\overset{(c)}{\leq} L_2\|\hat{\boldsymbol{\psi}}^k - \boldsymbol{\psi}^{*k}\|_2 + L_1\big(\|\breve{\boldsymbol{\theta}}_t^{k-1}(\hat{\boldsymbol{\Psi}}) - \breve{\boldsymbol{\theta}}_t^{k-1}(\boldsymbol{\Psi}^*)\|_2 +$$
$$\alpha\|\nabla_{\breve{\boldsymbol{\theta}}_t^{k-1}}\mathcal{L}(\breve{\boldsymbol{\theta}}_t^{k-1}(\hat{\boldsymbol{\Psi}}); \mathcal{D}_t^{\mathrm{trn}}) - \nabla_{\breve{\boldsymbol{\theta}}_t^{k-1}}\mathcal{L}(\breve{\boldsymbol{\theta}}_t^{k-1}(\boldsymbol{\Psi}^*); \mathcal{D}_t^{\mathrm{trn}})\|_2\big)$$

$$\leq L_2\|\hat{\boldsymbol{\psi}}^k - \boldsymbol{\psi}^{*k}\|_2 + L_1(1 + \alpha G_2)\|\breve{\boldsymbol{\theta}}_t^{k-1}(\hat{\boldsymbol{\Psi}}) - \breve{\boldsymbol{\theta}}_t^{k-1}(\boldsymbol{\Psi}^*)\|_2$$

$$\overset{(d)}{\leq} L_1(1 + \alpha G_2)\|\breve{\boldsymbol{\theta}}_t^{k-1}(\hat{\boldsymbol{\Psi}}) - \breve{\boldsymbol{\theta}}_t^{k-1}(\boldsymbol{\Psi}^*)\|_2 + L_2\|\hat{\boldsymbol{\Psi}} - \boldsymbol{\Psi}^*\|_1 \tag{55}$$

where $(a)$ follows from (7b), $(b)$ uses Assumption C.1, $(c)$ is from (7a) and Assumption A.2, and $(d)$ is due to that $\|\hat{\boldsymbol{\psi}}^k - \boldsymbol{\psi}^{*k}\|_2 \leq \max_{k=1}^K \|\hat{\boldsymbol{\psi}}^k - \boldsymbol{\psi}^{*k}\|_2 = \|\hat{\boldsymbol{\Psi}} - \boldsymbol{\Psi}^*\|_1$.

Solving the recursive relationship (55) using $\|\breve{\boldsymbol{\theta}}_t^k(\hat{\boldsymbol{\Psi}}) - \breve{\boldsymbol{\theta}}_t^k(\boldsymbol{\Psi}^*)\|_2 = 0$ gives

$$\big\|\breve{\boldsymbol{\theta}}_t^K(\hat{\boldsymbol{\Psi}}) - \breve{\boldsymbol{\theta}}_t^K(\boldsymbol{\Psi}^*)\big\|_2 \leq \begin{cases} \frac{1 - L_1^K(1+\alpha G_2)^K}{1 - L_1(1+\alpha G_2)} L_2\|\hat{\boldsymbol{\Psi}} - \boldsymbol{\Psi}^*\|_1, & \text{if } L_1(1+\alpha G_2) \neq 1 \\ KL_2\|\hat{\boldsymbol{\Psi}} - \boldsymbol{\Psi}^*\|_1, & \text{otherwise} \end{cases}, \tag{56}$$

which concludes that

$$\frac{1}{\sqrt{d}}\big\|\breve{\boldsymbol{\theta}}_t^K(\hat{\boldsymbol{\Psi}}) - \breve{\boldsymbol{\theta}}_t^K(\boldsymbol{\Psi}^*)\big\|_2 = \mathcal{O}\Big(\frac{1}{\sqrt{d}}\|\hat{\boldsymbol{\Psi}} - \boldsymbol{\Psi}^*\|_1\Big). \tag{57}$$

## D   Additional remarks regarding the theoretical results

Next, three important remarks regarding the derived error bounds will be provided.

*Remark* D.1 (Difference with convergence rate analysis). It is worth stressing these approximation error bounds are different from the convergence rate analysis. Essentially, it quantifies the impact of using a parametric $\breve{\mathrm{prox}}$ to approximate the optimal yet unknown $\overline{\mathrm{prox}}_{\mathcal{R}^*,\alpha}$. Moreover, although the bounds increase with $K$, it is important to note that $K$ is a sufficiently small constant (typically $1 \leq K \leq 5$) in the context of meta-learning (Finn et al., 2017), as the overall complexity for solving (2) scales linearly with $K$. Furthermore, the learning rate must satisfy $\alpha \in (0, 2/G_2)$ to guarantee a gradient-related descent direction, with $\alpha = 1/G_2$ being the optimal choice. A consequence of this choice is that $(1 + \alpha G_2)^K \in [2, 32]$, which ensures that the constant in the upper bounds will not diverge.

*Remark* D.2 (Factorability and scalability). Assumption 3.1 ensures that the prior dimension $D$ scales with the task-specific parameter dimension $d$. As $d$ in practice can be extremely large (e.g., $\Omega(10^5)$), a complete prior such as a full Gaussian pdf would incur prohibitively high complexity; that is, $D = \Theta(d^2) = \Omega(10^{10})$. A feasible simplification is to approximate the prior in $\mathbb{R}^d$ using the multiplication of $d$ pdfs in $\mathbb{R}$. This assumption essentially considers each dimension of $\boldsymbol{\theta}_t$ to be mutually independent, leading to $D = \Theta(d)$. Such an independence assumption is prevalent not only in meta-learning, but also in high-dimensional statistics when dealing with deep NNs.

*Remark* D.3 (Validity of learned proximal operator). The learnable $\check{\mathrm{prox}}$ in this paper remains a proximal operator, even when the corresponding regularizer is non-convex. Indeed, let $\check{\mathrm{prox}}^{-1}(\mathbf{x};\boldsymbol{\theta}) := \arg\min_{\mathbf{z}}\{\|\mathbf{z}\| \mid \check{\mathrm{prox}}(\mathbf{z};\boldsymbol{\theta}) = \mathbf{x}\}$ for $\mathbf{x} \in \{\check{\mathrm{prox}}(\mathbf{z};\boldsymbol{\theta})|A\mathbf{1}_d \preceq \mathbf{z} \preceq A\mathbf{1}_d\} := \mathcal{X}$, and $\mathbf{x}_0 := \check{\mathrm{prox}}(\mathbf{z}_0;\boldsymbol{\theta}) \in \mathcal{X}$. The stationary point condition of the proximal operator indicates $\check{\mathrm{prox}}^{-1}(\mathbf{x}_0;\boldsymbol{\theta}) - \mathbf{x}_0 \in \partial\mathcal{R}(\mathbf{x}_0;\boldsymbol{\theta})$; thus, one of the regularizers satisfying this condition is $\mathcal{R}(\boldsymbol{\theta}_t;\boldsymbol{\theta}) = \int_{\mathbf{x}\in\mathcal{X}:\mathbf{x}\prec\boldsymbol{\theta}_t}(\check{\mathrm{prox}}^{-1}(\mathbf{x};\boldsymbol{\theta}) - \mathbf{x})d\mathbf{x}$. Compared to non-expansive operators, proximal operators induced by non-convex regularizers have gained popularity in recent years thanks to their enhanced expressiveness, and their convergence guarantees (Hurault et al., 2022). Our method fails precisely within this category. Additionally, the PLFs should be monotone to qualify as a valid proximal operator; see e.g., (Gribonval & Nikolova, 2020, Theorem 1). In practice, the sought monotonicity can be established by enforcing $\psi_{i,c}^k \leq \psi_{i,c'}^k$, $\forall c \leq c'$. This can be readily achieved upon defining $\psi_{i,c+1}^k := \psi_{i,c}^k + \exp(\Delta\psi_{i,c}^k)$, $\forall i, c, k$, and then learn the log-increment $\{\Delta\psi_{i,c}^k\}_{i,c,k}$. Interestingly, we have observed that the learned PLFs are exactly monotone functions (see e.g., Fig. 4), even without an explicit constraint. This observation suggests an inherent preference for monotonic PLFs by the data.

# E   DETAILED SETUPS OF NUMERICAL TESTS

In this section, we introduce the dataset and elaborate the detailed setups of the numerical tests.

The miniImageNet dataset (Vinyals et al., 2016) consists of $60,000$ natural images sampled from the full ImageNet (ILSVRC-12) dataset. These images are categorized into 100 classes, each with 600 labeled samples. As suggested by (Ravi & Larochelle, 2017), all images are cropped and resized to size $84 \times 84$. The dataset is split into 3 disjoint groups containing 64, 16 and 20 classes, which can be respectively accessed during the training, validation, and testing phases of meta-learning. The experimental setups follow from the standard $M$-class $N$-shot few-shot learning protocol (Ravi & Larochelle, 2017; Finn et al., 2017). Specifically, $\mathcal{D}_t^{\mathrm{trn}}$ per task $t$ includes $M$ classes randomly drawn from the dataset, each containing $N$ labeled data. As a result, it is clear that $|\mathcal{D}_t^{\mathrm{trn}}| = MN$ for each $t$.

The TieredImageNet (Ren et al., 2018) dataset is a larger subset of the ImageNet dataset, composed of $779,165$ images from 608 classes. Likewise, all the images are preprocessed to have size $84 \times 84$. Instead of using a random split, classes are partitioned into 34 categories according to the hierarchy of ImageNet dataset. Each category contains 10 to 30 classes. These categories are further grouped into 3 different sets: 20 for training, 6 for validation, and 4 for testing.

We utilized the group of hyperparameters described in MAML (Finn et al., 2017) consistently throughout all the tests. To be specific, the maximum number $K$ of PGD steps (7) is 5, and the total number $R$ of mini-batch SGD iterations (8) is $60,000$. The number of convolutional channels is 64 for MetaProxNet+MAML, and 128 for MetaProxNet+MC. The learning rates for PGD and SGD are $\alpha = 0.01$ and $\beta = 0.001$, with batch size $B = 4$. Adam optimizer is employed for tieredImageNet, while SGD with Nesterov momentum of 0.9 and weight decay of $10^{-4}$ is used for miniImageNet.

The interval $[-A, A]$ and number $C$ of pieces are determined through a grid search leveraging the validation tasks. For both miniImageNet and TieredImgeNet datasets, $A = 0.02$ and $C = 5$. We found that $C = 5$ suffices to reach a satisfactory performance, while larger $C$ only contributes marginally to MetaProxNet's empirical performance. This suggests the constants hidden inside the error bounds $\mathcal{O}(1/C)$ and $\mathcal{O}(1/C^2)$ can be sufficiently small in practice.

# F   COMPLEXITY ANALYSIS OF METAPROXNET

It can be observed from (6) and (7b) that the per-step piecewise linear function (PLF) $\check{\mathrm{prox}}^k$ applies a dimension-wise *affine transformation* to its input. Although the per-dimension $\check{\mathrm{prox}}_i^k$ consists of $C + 1$ parameters, the affine transformation merely relies on *a single piece* $[\zeta_{i,c-1}^k, \zeta_{i,c}^k]$ of the PLF. Thus, each computation involves only two control points $\psi_{i,c-1}^k, \psi_{i,c}^k$ for every $k = 1, \dots, K$ and $i = 1, \dots, d$. As a result, the forward calculation and backward differentiation of PLFs both incur complexity $\mathcal{O}(Kd)$. While the $K$-step GD of (7a) also exhibits forward and backward complexities of $\mathcal{O}(Kd)$, its constant hidden within $\mathcal{O}$ can be much larger, as the convolutional operations in the

Table 3: Comparison of (normalized) running time on the 5-class 5-shot miniImageNet dataset.

| Method | Forward | Backward | Total |
|---|---|---|---|
| MAML | ×0.182 | ×0.818 | ×1 (reference) |
| MetaProxNet+MAML | ×0.202 | ×0.836 | ×1.038 |
| MetaCurvature | ×0.188 | ×0.834 | ×1.022 |
| MetaProxNet+MetaCurvature | ×0.208 | ×0.863 | ×1.071 |

Table 4: Ablation test regarding weight untying using a 4-layer 32-channel CNN.

| Method | 5-class | | 10-class | |
| | 1-shot (%) | 5-shot (%) | 1-shot (%) | 5-shot (%) |
|---|---|---|---|---|
| MAML | $48.70_{\pm 1.84}$ | $63.11_{\pm 0.92}$ | $31.27_{\pm 1.15}$ | $46.92_{\pm 1.25}$ |
| MetaProxNet (shared $\breve{\mathrm{prox}}$) + MAML | $51.16_{\pm 1.45}$ | $67.08_{\pm 0.71}$ | $\mathbf{34.62}_{\pm \mathbf{0.93}}$ | $50.26_{\pm 0.48}$ |
| MetaProxNet (per-step $\breve{\mathrm{prox}}_k$) + MAML | $\mathbf{53.58}_{\pm \mathbf{1.43}}$ | $\mathbf{67.88}_{\pm \mathbf{0.72}}$ | $34.80_{\pm 0.91}$ | $\mathbf{51.03}_{\pm \mathbf{0.51}}$ |

CNN are more time-consuming than the affine ones. Consequently, PLFs contribute only marginally ($< 5\%$) to the overall complexity compared to their backbone. This is also evidenced numerically in Tab. 3. Moreover, Assumption (3.1) indeed ensures that the prior dimension $D$ scales with $d$, thereby mitigating any substantial complexity increase. Specifically, in practical scenarios where $d$ is extremely large (e.g., $d = 121,093$ or $463,365$ in our experiments), employing a complete prior, such as a full Gaussian pdf, would yield $D = \Theta(d^2)$. A feasible simplification is to approximate the prior in $\mathbb{R}^d$ using the multiplication of $d$ pdfs in $\mathbb{R}$, which leads to $D = \Theta(d)$.

## G  EXTRA ABLATION TESTS REGARDING WEIGHT UNTYING

The next test examines the effectiveness of the weight-untying technique; i.e., the per-step $\breve{\mathrm{prox}}$. The experiment is conducted on the miniImageNet dataset with a 4-layer 32-channel CNN, and the corresponding results are summarized in Tab. 4. It is seen that the per-step proximal operator consistently outperforms the shared one in all four tests. In fact, the per-step $\breve{\mathrm{prox}}_k$ inherently corresponds to an adaptive prior, which evolves with the optimization process. The same technique was originally provided by the renowned LISTA algorithm (Gregor & LeCun, 2010), which pioneered algorithm unrolling, and has since been widely adopted by the community on inverse problems.

## H  NUMERICAL VERIFICATION OF ERROR BOUNDS

Next, a toy numerical test is carried out to verify the derived PGD error bounds. For simplicity, we will exclusively focus on Theorem 3.2, while similar analysis can be readily applied to Theorem 3.3. Consider tasks defined by the linear relationship $y_t^n = \mathbf{w}_t^{*\top} \mathbf{x}_t^n + e_t^n$, and a linear prediction model $\hat{y}_t^n = f(\mathbf{x}_t^n; \boldsymbol{\theta}_t) := \boldsymbol{\theta}_t^\top \mathbf{x}_t^n$ with squared $\ell_2$ loss $\mathcal{L}(\boldsymbol{\theta}_t; \mathcal{D}_t^{\mathrm{trn}}) := \frac{1}{2} \sum_{n=1}^{N_t^{\mathrm{trn}}} ||y_t^n - \boldsymbol{\theta}_t^\top \mathbf{x}_t^n||_2^2$, where the

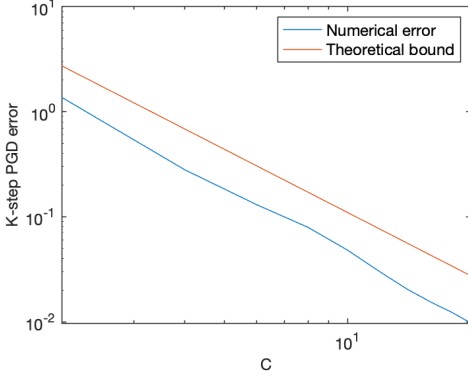

Figure 5: Comparison of theoretical error bound (9) and numerical error in linear regression.

unknown oracle $\mathbf{w}_t^* \sim \text{Uniform}([-3,3]^d)$ and $\mathbf{x}_t^n \sim \mathcal{N}(\mathbf{0}_d, \mathbf{I}_d)$. In the test, we set $K = 5, \alpha = 0.01, d = 64, |\mathcal{D}_t^{\text{trn}}| = 8, \boldsymbol{\theta}^{\text{init}} = \mathbf{0}_d, A = 3$ and $C$ varying from 2 to 20. The target optimal proximal operator to be approximated is defined in (27) with $G_1 = 1$. Additionally, the Lipschitz constant $G_2$ of $\nabla \mathcal{L}(\boldsymbol{\theta}_t; \mathcal{D}_t^{\text{trn}})$ is numerically computed from the randomly generated training data matrix $\mathbf{X}_t^{\text{trn}}$.

The plot comparing the numerical PGD error and its upper bound can be found in Fig. 5. It is observed that the numerical error aligns with the theoretical bound up to a small constant in the log-scale. This discrepancy arises because the upper bound considers the worst-case scenario, where the largest error between $\text{prox}_{\mathcal{R}^*,\alpha}$ and $\text{prox}^k$ is reached at each PGD step. Furthermore, this constant gap suggests that the numerical error is in the same order with the bound (notice that both axes are in log-scale); that is, $\mathcal{O}(\frac{1}{C^2})$. This empirically corroborates our theoretical proofs.

## I  CASE STUDY: FEW-SHOT REGRESSION

Next, a straightforward yet illuminating numerical case study is provided to show the claimed superior prior expressiveness. Consider few-shot regression tasks defined by the linear data model $y_t^n = \mathbf{w}_t^{*\top} \mathbf{x}_t^n + e_t^n$, where the unknown per-task weights $\{\mathbf{w}_t^*\}_{t=1}^T$ are i.i.d. samples from the oracle pdf $p(\mathbf{w}^*) := \frac{1}{2}\text{Uniform}([-11,-10]^d) + \frac{1}{2}\text{Uniform}([10,11]^d)$, and $e_t^n \sim \mathcal{N}(0, \sigma_e^2), \forall t, n$ is the additive white Gaussian noise. Further, consider a linear prediction model $\hat{y}_t^n = f(\mathbf{x}_t^n; \boldsymbol{\theta}_t) := \boldsymbol{\theta}_t^\top \mathbf{x}_t^n$. Since $p(\mathbf{w}^*)$ is symmetric and isotropic, the optimal Gaussian prior of $\boldsymbol{\theta}_t$ for this case must have a mean of $\mathbf{0}_d$ an isotropic covariance. In other words, if the prior pdf is chosen to be Gaussian $p(\boldsymbol{\theta}_t; \boldsymbol{\theta}) = \mathcal{N}(\boldsymbol{\mu}, \boldsymbol{\Sigma}), \boldsymbol{\theta} := [\boldsymbol{\mu}^\top, \text{vec}(\boldsymbol{\Sigma})^\top]^\top$, its optimal parameter $\boldsymbol{\theta}^*$ must consist of $\boldsymbol{\mu}^* = \mathbf{0}_d$ and $\boldsymbol{\Sigma}^* = \lambda^{*-1}\mathbf{I}_d$ for some $\lambda^* \in \mathbb{R}$. As a result, the corresponding regularizer for this prior is $\mathcal{R}(\boldsymbol{\theta}_t; \boldsymbol{\theta}^*) = \frac{\lambda^*}{2}\|\boldsymbol{\theta}_t\|_2^2$, which prevents $\boldsymbol{\theta}_t$ deviating far from $\mathbf{0}_d$. However, the ground-truth task model implies that the optimal $\boldsymbol{\theta}_t^*$ should belong to the set $\mathcal{S} := [-11,-10]^d \cup [10,11]^d$, and the regularizer is thus a barrier for optimizing $\boldsymbol{\theta}_t$. In contrast, if the prior pdf is allowed to be non-Gaussian, the optimal prior will be the ground-truth one, i.e., $p(\boldsymbol{\theta}_t; \boldsymbol{\theta}^*) = \text{Uniform}(\mathcal{S})$. The corresponding proximal operator in this case is the projection $\text{prox}_{\mathcal{R}^*,\alpha}(\mathbf{z}) = \mathbb{P}_{\mathcal{S}}(\mathbf{z})$, which is exactly a piecewise linear function driving $\boldsymbol{\theta}_t$ to the oracle set. In summary, the Gaussian pdf fails to match the underlying prior due to its inherent unimodality, while the more expressive PLF-induced prior can perfectly align with the groundtruth prior to enhance the task-level learning.

For visualization purpose, a numerical test is carried out with $d = 1$. The remaining parameters are $|\mathcal{D}_t^{\text{trn}}| = 2, |\mathcal{D}_t^{\text{val}}| = 5, \sigma_e = 1, K = 5, R = 10,000, \alpha = 0.1, \beta = 0.01, A = 15, C = 30, B = 4$ and $x_t^n \sim \mathcal{N}(0,1)$. In Fig. 6, the linear function learned by MAML is inclined to have a slope close to 0, while the PLFs in MetaProxNet quickly refine $\theta_t$ into the set $\mathcal{S} = [-11,-10] \cup [10, 11]$. This empirical observation substantiates the advocated superior prior expressiveness.

In practical tasks such as drug discovery and robotic manipulations, the oracle model parameters can have similar multi-modal pdf defined on a bounded set. In drug discovery for instance, the efficacy of a drug might only manifest when one component accounts for a specific portion.

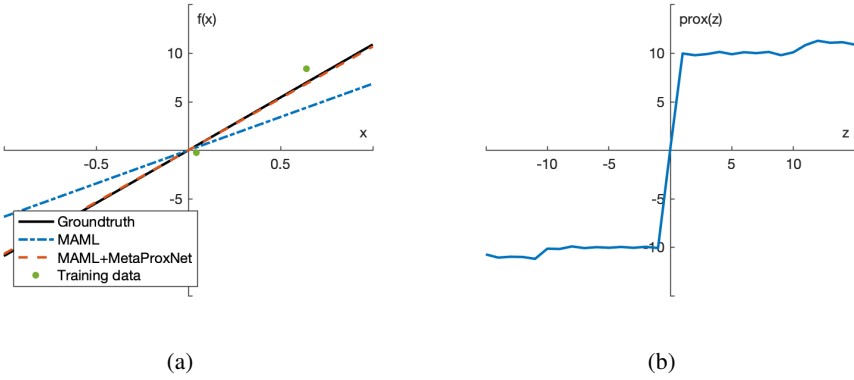

(a)                                      (b)

Figure 6: Visualization of 1-d few-shot regression case study; (a) comparison of MAML and MAML+MetaProxNet; (b) the learned PLF-based proximal operator averaged across PGD steps.

## J   A BRIEF INTRODUCTION TO ALGORITHM UNROLLING

Algorithm unrolling was first introduced in (Gregor & LeCun, 2010) to solve the inverse problem. In particular, it aims to recover a (transformed) signal $\mathbf{x} \in \mathbb{R}^n$ from its compressed measurements

$$\mathbf{y} = \mathbf{A}\mathbf{x} + \mathbf{e} \tag{58}$$

where $\mathbf{A} \in \mathbb{R}^{m \times n}$ is a given matrix with $m \ll n$, and $\mathbf{e}$ is additive white Gaussian noise. Since the system (58) is under-determined, it has infinitely many solutions. To ensure the uniqueness of the solution, a prudent remedy is to rely on the prior $p(\mathbf{x})$, which yields

$$\hat{\mathbf{x}} = \operatorname*{argmin}_{\mathbf{x}} \|\mathbf{y} - \mathbf{A}\mathbf{x}\|_2^2 + \mathcal{R}(\mathbf{x}). \tag{59}$$

In the above, $\|\mathbf{y} - \mathbf{A}\mathbf{x}\|_2^2$ and $\mathcal{R}(\mathbf{x})$ correspond to the nll $-\log p(\mathbf{y}; \mathbf{x})$ and nlp $-\log p(\mathbf{x})$, respectively. As nature signals are inherently sparse in certain transform domains such as Fourier and wavelet ones, a popular choice is the sparse prior with $\mathcal{R}(\mathbf{x}) = \lambda \|\mathbf{x}\|_1$. With such a prior, the resultant optimization problem (59) can be efficiently solved by the well-documented iterative soft-thresholding algorithm (ISTA), which involves a two-step update rule

$$\mathbf{z}^k = \mathbf{x}^{k-1} - \alpha \mathbf{A}^\top (\mathbf{A}\mathbf{x}^{k-1} - \mathbf{y}) = (\mathbf{I}_n - \alpha \mathbf{A}^\top \mathbf{A})\mathbf{x}^{k-1} + \alpha \mathbf{A}^\top \mathbf{y} \tag{60a}$$

$$\mathbf{x}^k = \operatorname*{argmin}_{\mathbf{x}} \frac{1}{2\alpha} \|\mathbf{z}^k - \mathbf{x}\|_2^2 + \|\mathbf{x}\|_1 = \mathbb{S}_{\alpha\lambda}(\mathbf{z}^k), \; k = 1, \ldots, K \tag{60b}$$

Here, $\mathbb{S}_{\alpha\lambda}$ is the soft-thresholding operator shown in Fig. 2b.

When given a dataset $\{(\mathbf{x}_n, \mathbf{y}_n)\}_{n=1}^N$, it is possible to enhance the accuracy of $\hat{\mathbf{x}}$ by learning a more efficient optimization rule. In (Gregor & LeCun, 2010), the two steps (60) of ISTA for each $k = 1, \ldots, K$ are replaces by two learnable NN blocks

$$\mathbf{z}^k = \mathbf{W}_{\mathbf{x}}^k \mathbf{x}^{k-1} + \mathbf{W}_{\mathbf{y}}^k \mathbf{y} \tag{61a}$$

$$\mathbf{x}^k = \mathbb{S}_{\beta^k}(\mathbf{z}^k) \tag{61b}$$

with $\boldsymbol{\theta} := \{(\mathbf{W}_{\mathbf{x}}^k, \mathbf{W}_{\mathbf{y}}^k, \beta^k)\}_{k=1}^K$ being the learnable weights of the NN. Denoting by $f(\mathbf{y}; \boldsymbol{\theta})$ this multi-block NN mapping, the NN-based update rule can be learned via

$$\min_{\boldsymbol{\theta}} \sum_{n=1}^N \|\mathbf{x}_n - f(\mathbf{y}_n; \boldsymbol{\theta})\|_2^2. \tag{62}$$

This method of unfolding and substituting the iterations of an optimization algorithm to form a multi-block NN is known as *algorithm unrolling*, and the resultant NN is termed an *unrolled NN*.

## K   RELATED WORK

**NN-based meta-learning:**   Recurrent neural network (RNN) has been introduced in (Hochreiter et al., 2001) to learn the update rule of the task-specific model parameters $\boldsymbol{\theta}_t$, with prior encoded in the RNN's weights. Following this work, different RNN architechtures have been explored to enhance the learning of the update rules. On the one hand, gradient information has been leveraged in (Andrychowicz et al., 2016; Li & Malik, 2017; Ravi & Larochelle, 2017) to mimic the gradient-based optimization. On the other hand, temporal convolutions and soft attention have been utilized to aggregate and pinpoint information from past experiences (Mishra et al., 2018). More recently, this paradigm of NN-based optimization has been extended to Beyesian meta-learning, aiming to infer the posterior pdf $p(\boldsymbol{\theta}_t | \mathbf{y}_t^{\text{trn}}; \mathbf{X}_t^{\text{trn}})$ (Gordon et al., 2019). Due to the blackbox nature of the RNNs however, it is hard to interpret the impact of learned prior from the NN-based update rules.

**Optimization-based meta-learning:**   To empower fast adaptation to a new task, MAML capitalized on learning a task-invariant initialization (Finn et al., 2017), with task-level learning defined by a cascade of a few GD steps on the task-specific parameter $\boldsymbol{\theta}_t$. Intuitively, by descending a small number of steps, $\boldsymbol{\theta}_t$ should not deviate too far away from its initial value $\boldsymbol{\theta}$. In fact, it has been pointed out in (Grant et al., 2018) that MAML's GD solver satisfies $\boldsymbol{\theta}_t^*(\boldsymbol{\theta}) \approx \hat{\boldsymbol{\theta}}_t(\boldsymbol{\theta}) = \operatorname{argmin}_{\boldsymbol{\theta}_t} \mathcal{L}(\boldsymbol{\theta}_t; \mathcal{D}_t^{\text{trn}}) + \frac{1}{2} \|\boldsymbol{\theta}_t - \boldsymbol{\theta}\|_{\boldsymbol{\Lambda}_t}^2$, where $\boldsymbol{\Lambda}_t$ is determined by hyperparamters of GD. This observation indicates that MAML's optimization strategy approximates an implicit Gaussian prior $p(\boldsymbol{\theta}_t; \boldsymbol{\theta}) = \mathcal{N}(\boldsymbol{\theta}, \boldsymbol{\Lambda}_t^{-1})$, with initialization $\boldsymbol{\theta}^{\text{init}} = \boldsymbol{\theta}$ serving as the mean vector. Following MAML,

a spectrum of algorithms have been developed to encode different priors. For instance, Meta-SGD (Li et al., 2017) augments MAML by meta-learning a dimension-wise step size, which essentially corresponds to a per-step diagonal Gaussian prior. Other examples of the induced priors include isotropic Gaussian (Rajeswaran et al., 2019; Abbas et al., 2022), diagonal Gaussian (Ravi & Beatson, 2019; Nguyen et al., 2020), per-step block-diagonal Gaussian (Park & Oliva, 2019; Flennerhag et al., 2020), and implicit Gaussian (Baik et al., 2020; 2021).

Another line of research termed metric-based meta-learning (which can be either NN-based or optimization-based) splits the model into an embedding "body" and a classifier/regressor "head," and learn their priors independently (Vinyals et al., 2016; Snell et al., 2017; Sung et al., 2018; Li et al., 2020a). In particular, with $\boldsymbol{\theta}_t^{\mathrm{body}}$ and $\boldsymbol{\theta}_t^{\mathrm{head}}$ denoting the corresponding partitions of $\boldsymbol{\theta}_t$, the prior is presumed factorable as $p(\boldsymbol{\theta}_t; \boldsymbol{\theta}) = p(\boldsymbol{\theta}_t^{\mathrm{body}}; \boldsymbol{\theta}^{\mathrm{body}}) p(\boldsymbol{\theta}_t^{\mathrm{head}}; \boldsymbol{\theta}^{\mathrm{head}})$, $\boldsymbol{\theta} = [\boldsymbol{\theta}^{\mathrm{body}\top}, \boldsymbol{\theta}^{\mathrm{head}\top}]^\top$. On the one hand, the head typically has a nontrivial prior such as the Gaussian one (Bertinetto et al., 2019; Lee et al., 2019). On the other hand, the body's prior is intentionally restricted to a degenerate pdf $p(\boldsymbol{\theta}_t^{\mathrm{body}}; \boldsymbol{\theta}^{\mathrm{body}}) := \delta(\boldsymbol{\theta}_t^{\mathrm{body}} - \boldsymbol{\theta}^{\mathrm{body}})$, where $\delta(\cdot)$ is the Dirac delta function. Although freezing the body in task-level optimization remarkably reduces its complexity, it often leads to degraded performance compared to the full GD update (Raghu et al., 2020). In additional to degenerate priors, sparse priors have also been recently investigated to selectively update a subset of parameters (Lee & Choi, 2018; Tian et al., 2020a).

Compared to these preset prior pdfs of fixed shapes, the focus of this work is to learn a data-driven prior pdf that can dynamically adjust itself to fit the given tasks.

**Algorithm unrolling:** The advocated MetaProxNet pertains to the algorithm unrolling category (Gregor & LeCun, 2010; Monga et al., 2021; Li et al., 2020b). Closely related to our work is the deep regularization approach introduced in (Li et al., 2020a). This method shares similar high-level idea of incorporating priors into PGD optimization iterations through algorithm unrolling. In (Li et al., 2020a), the hidden representations are transformed into a domain conducive to easy regularization by a predefined prior pdf (e.g., isotropic Gaussian). In contrast, our MetaProxNet approach involves the direct learning of the proximal operator within the parametric space.

