# OpenReview forum: "Meta-Learning Priors Using Unrolled Proximal Networks"
_ICLR.cc/2024/Conference — ICLR 2024 poster_

### Official Review · Reviewer_YbkU · 2023-10-27

**Soundness:** 4 excellent
**Presentation:** 4 excellent
**Contribution:** 3 good
**Rating:** 8
**Confidence:** 2

**Summary:**

The paper considers optimization-based meta-learning algorithms, i.e., algorithms that employ a bi-level optimization strategy to optimize task-specific and task-global parameters. This approach can be interpreted as the task-specific parameters being regularized towards a task-global prior optimized in the outer loop. The authors propose a novel method to induce more general/expressive priors in comparison to related work, by learning proximal operators for proximal gradient descent using unrolled NNs. The authors motivate a piecewise linear parametrization of this proximal operator, derive an algorithm to optimize the corresponding parameters, and analyze error bounds. They compare their approach against a range of optimization-based meta-learning algorithms.

**Strengths:**

The paper is well-written and both the theoretical exposition as well as the experimental evaluation seem to be well fleshed out. Algorithmic choices such as using piecewise linear approximations for the proximal operator appear well-motivated. The experiments contain comparisons to a wide range of optimization-based meta-learning algorithms, demonstrating superior performance of the proposed method on the miniImageNet benchmark. Further ablations underline the effectiveness of algorithmic design choices. In summary, the paper appears as an interesting and effective approach to derive priors for optimization-based meta-learning.

Therefore, I vote for acceptance. However, my recommendation is with low confidence, so I might decrease my score during the rebuttal period.

**Weaknesses:**

The paper could be further strengthend by providing experimental results on benchmarks other than miniImageNet. Furthermore, I would appreciate a discussion of how the method relates to and/or could be extended to fully Bayesian approaches like Probabilistic MAML [1] or Bayesian MAML [2].

[1] Kim et al., Bayesian Model-Agnostic Meta-Learning, NeurIPS 2018

[2] Finn et al., Probabilistic Model-Agnostic Meta-Learning, NeurIPS 2018

**Questions:**

cf. weaknesses

---

> ### Author Response · Authors · 2023-11-13
>
> Thank you for the interest in this work, and for the valuable questions. The two concerns regarding extended numerical tests and generalization to Bayesian meta-learning approaches will be answered below.
>
> **1. Extended numerical tests**
> While our current numerical results are evaluated on miniImageNet and tieredImageNet, we are writing codes to implement MetaProx on the [Caltech-UCSD Birds (CUB)-200-2011 dataset](https://www.vision.caltech.edu/datasets/cub_200_2011/) and [Meta-Dataset](https://github.com/google-research/meta-dataset). In addition, experiments on reinforcement learning tasks has also been prioritized in our research agenda.
>
> **2. Generalization to Bayesian meta-learning**
> The objective of Bayesian meta-learning takes into account the uncertainties in the model parameter $\boldsymbol{\theta}_t$, which incurs a similar bilevel objective
>
> $$
> \max_{\boldsymbol{\theta}} \prod_{t=1}^T p (\mathbf{y}_t^\mathrm{val} | \mathcal{D}_t^{\mathrm{trn}}; \mathbf{X}_t^\mathrm{val}, \boldsymbol{\theta}) = \prod_t \int p (\mathbf{y}_t^\mathrm{val} | \boldsymbol{\theta}_t; \mathbf{X}_t^\mathrm{val}) p(\boldsymbol{\theta}_t |  \mathbf{y}_t^{\mathrm{trn}}; \mathbf{X}_t^{\mathrm{trn}}, \boldsymbol{\theta}) d\boldsymbol{\theta}_t
> $$
> $$
> \text{s.t.} ~p(\boldsymbol{\theta}_t |  \mathbf{y}_t^{\mathrm{trn}}; \mathbf{X}_t^{\mathrm{trn}}, \boldsymbol{\theta}) \propto p(\mathbf{y}_t^\mathrm{trn} | \boldsymbol{\theta}_t; \mathbf{X}_t^\mathrm{trn}) p(\boldsymbol{\theta}_t; \boldsymbol{\theta}), ~\forall t.
> $$
>
> Akin to its deterministic counterpart, the exact posterior pdf $p(\boldsymbol{\theta}_t |  \mathbf{y}_t^{\mathrm{trn}}; \mathbf{X}_t^{\mathrm{trn}}, \boldsymbol{\theta})$ is generally intractable. A practical alternative is to rely on approximate inference techniques. In Bayesian MAML (BMAML), the inference is carried out through particle sampling, where the complexity scales in linear with the number of particles. In other Bayesian meta-learning approaches such as Probabilistic MAML, Amortized Bayesian Meta-Learning (ABML) [1] and VAMPIRE [2], the posterior is approximated through variational inference (VI). In particular, VI seeks for a tractable surrogate pdf $p(\boldsymbol{\theta}_t;\mathbf{v})$ that best estimates $p(\boldsymbol{\theta}_t |  \mathbf{y}_t^{\mathrm{trn}}; \mathbf{X}_t^{\mathrm{trn}}, \boldsymbol{\theta})$ by minimizing their KL divergence
> $$
> \mathbf{v}_t^* = \mathop{\arg\min}_v~\mathrm{KL} \big(p (\boldsymbol{\theta}_t; \mathbf{v}) || p(\boldsymbol{\theta}_t |  \mathbf{y}_t^{\mathrm{trn}}; \mathbf{X}_t^{\mathrm{trn}}, \boldsymbol{\theta})\big)
> $$
>
> which is equivalent to the maximization of the so-termed evidence lower bound (ELBO)
> $$
> \mathbf{v}_t^*=\mathop{\arg\max}_v~\mathbb{E}_p{\scriptstyle (\boldsymbol{\theta}_t; \mathbf{v})} [ \log p(\mathbf{y}_t^{\mathrm{trn}} | \boldsymbol{\theta}_t;\mathbf{X}_t^{\mathrm{trn}})] - \mathrm{KL} \big( p(\boldsymbol{\theta}_t; \mathbf{v}) || p(\boldsymbol{\theta}_t; \boldsymbol{\theta})\big).
> $$
>
> Upon defining loss $\mathcal{L}(\mathbf{v};\mathcal{D}_t^\mathrm{trn}):=\mathbb{E}_p{\scriptstyle (\boldsymbol{\theta}_t; \mathbf{v})} [ -\log p(\mathbf{y}_t^{\mathrm{trn}} | \boldsymbol{\theta}_t;\mathbf{X}_t^{\mathrm{trn}})]$ to be the expected nll, and regularizer $\mathcal{R}(\mathbf{v};\boldsymbol{\theta}):=\mathrm{KL} \big( p(\boldsymbol{\theta}_t;\mathbf{v}) || p(\boldsymbol{\theta}_t; \boldsymbol{\theta})\big)$, the VI objective reduces to
> $$
> \mathbf{v}_t^*=\mathop{\arg\min}_v~\mathcal{L}(\mathbf{v};\mathcal{D}_t^\mathrm{trn})+\mathcal{R}(\mathbf{v};\boldsymbol{\theta})
> $$
> which is exactly (1b) of our manuscript. In fact, the regularizer in VI can be any function assessing the divergence between two pdf. Similar to the deterministic case where Gaussian prior is popular, most existing Bayesian meta-learning algorithms select KL divergence as the regularizer, which is known to be unreliable in certain cases (e.g., when two pdfs have disjoint supports). Moreover, since the ELBO is typically optimized using gradient ascent in Bayesian meta-learning, MetaProx can be readily incorporated into VI to learn a data-driven divergence function.
>
> [1] S. Ravi, and A. Beatson, "Amortized bayesian meta-learning," ICLR 2018.
>
> [2] C. Nguyen et al., “Uncertainty in Model-Agnostic Meta-Learning using Variational Inference,” WACV 2020.

---

> > ### Comment · Reviewer_YbkU · 2023-11-22
> > **Thanks for the answer**
> >
> > Thank you for your answer and for providing further insights. After reading the other reviews, I still think the paper makes a solid contribution. Therefore, I keep my score.

---

### Official Review · Reviewer_W1mq · 2023-10-29

**Soundness:** 3 good
**Presentation:** 3 good
**Contribution:** 3 good
**Rating:** 8
**Confidence:** 3

**Summary:**

This work presents a novel approach to meta-learning dubbed MetaProx, which leverages unrolled proximal neural networks to learn flexible and interpretable priors. The core concepts of MetaProx involve decompose the proximal gradient descent algorithm into a multi-block neural network, with each block comprising a data consistency module and a trainable piecewise linear function that approximates the proximal operator.

Theoretical analysis of this approximation demonstrates its capability to effectively model smooth and non-smooth proximal functions. Empirical evaluations of MetaProx in the context of few-shot classification tasks reveal its superior performance compared to other SOTA meta-learning techniques employing either RNN-base or handcrafted priors.

**Strengths:**

1. The idea of learning priors through unrolled PGD is a novel and intriguing approach. This method facilitate the model to learn a broader range of complex and adaptable priors in a data-driven manner, thanks to the inclusion of non-smooth and even discontinuous functions in the regularization term of the proximal operator. Furthermore, it enhances both model expressiveness and interpretability when compared to traditional meta-learning approaches that rely on predefined priors. For example, in optimization-based meta-learning methods like LEO and ANIL, an explicit Gaussian or Laplace prior is imposed on model parameters during inner-loop adaptation. To maintain computational efficiency and feasibility, the authors employ piecewise linear functions to approximate the proximal operator. Additionally, they further reduce costs by employing fixed discretization points for the piecewise linear functions instead of simultaneous learning them.
2. The paper provides a rigorous theoretical analysis bounding the approximation errors, offering valuable practical guidelines for hyper-parameter tuning, like C.
3. Comprehensive empirical evaluation on benchmark datasets and comparisons with state-of-the-art meta-learning methods. The consistent performance gains validate the effectiveness of MetaProx.

**Weaknesses:**

While this paper presents a solid contribution with rigorous analysis and experimental validation on few-shot classification, the scope of applications could be expanded to further demonstrate the generality of the proposed MetaProx method. In particular, the current empirical evaluation is limited to few-shot classification tasks. To reveal the full merits of MetaProx and its ability to learn flexible priors, additional experiments on other few-shot learning domains would make the work stronger. For example, further testing MetaProx on few-shot regression tasks, few-shot policy learning in reinforcement learning, or few-shot time series forecasting would verify its effectiveness beyond classification.

The paper currently visualizes the learned proximal operators averaged across iterations. An insightful extension would be to visualize the evolution of the learned priors at each unrolled PGD step, if possible. Analyzing how the proximal functions adapt over the course of optimization could provide deeper understanding into the dynamics of MetaProx and the induced priors.

Furthermore, the work by [1] on proximal layers for deep regularization is relevant, as it shares the high-level idea of encoding priors into optimization-based neural network training. A key difference is that [1] transforms hidden representations to comply with preset priors, while this work learns proximal operators over model parameters. Comparing and contrasting with [1] could potentially help situate the contributions of this paper.

[1]. Mao Li, Yingyi Ma, and Xinhua Zhang. Proximal mapping for deep regularization. Advances in Neural Information Processing Systems 2020.

**Questions:**

1. How do you anticipate the performance of MetaProx on few-shot regression or reinforcement learning tasks? Is it ready to be extend to these domains? Are there any challenges to be addressed?
2. Do you have any ideas to help interpret the high-dimensional learned prior, beyond visualizing the 1D components? Would it be possible to visual the evolution of learned prior after each unrolled PGD steps?
3. Does MetaProx require careful initialization or scheduling of $\lambda$ to avoid instability during training?
4. The method mainly evaluated on 5-way 1/5-shot classification benchmarks. How do you expect MetaProx to perform on more challenging few-shot settings, like 20-way 1/5-shot?
5. Is there trade-off between the regularization term and the loss function in eq (1b)?

---

> ### Author Response · Authors · 2023-11-13
>
> We appreciate your interest in this work and the constructive feedback provided. The issues raised are addressed one-by-one next.
>
> **1. Response to Weakness 1 & Question 1**
> As a flexible plug-in module, our MetaProx can be readily extended to few-shot regression and reinforcement learning (RL) tasks. Remarkably, the performance improvement by MetaProx mainly attributes to the enhanced prior expressiveness. Hence, it is expected to be particularly effective for tasks that exhibit intricate data distributions and require complex priors. To support this claim, a numerical case study on few-shot regression tasks will be provided in Appendix I of the updated manuscript.
> In addition, implementing MetaProx in RL applications such as few-shot policy learning is also an intriguing direction which has been prioritized in our research agenda. One noteworthy challenge towards this direction is known as “meta-overfitting”, wherein an overly complex prior can overfit when the number $T$ of given tasks is also limited. While this challenge can be alleviated by employing a small $C$, Theorems 3.2 and 3.3 indicate a potential increase of approximation error in such case. Therefore, a potential remedy is to explore a proximal operator with improved error bounds.
>
> **2. Response to Weakness 2 & Question 2**
> The evolution of the prior in each PGD step $k=1,\ldots,5$ has been visualized in Figure 4, marked in different colors. It is seen that the acquired PLFs keep shallow layer weights being sparse around the initial values (i.e., less updated with a soft-thresholding) when $k$ is small, while deep layers can be updated freely along its gradient directions. In other words, the learned prior freezes the weights of the embedding function and exclusively train the model head in the initial iterations. Once the head has attained sufficiently training with the increasing of $k$, the prior gradually unfreeze the shallow layers and proceed with optimizing the entire model. In fact, this learned update strategy closely resembles the widely adopted “gradual unfreezing” training approach for fine-tuning large-scale models, which has been proven effective in various practical applications of [computer vision](https://www.tensorflow.org/guide/keras/transfer_learning) and [natural language processing](https://aclanthology.org/P18-1031.pdf).
>
> **3. Response to Weakness 3**
> Thank you for highlighting the parallels and distinctions between MetaProx and [1]. The missing reference and comparison will be added to the revised manuscript.
>
> **4. Response to Question 3**
> The PLFs in MetaProx can be initialized according to the prior knowledge about the optimization process. In particular, we set initially $\psi_{i,c}^k = \zeta_{i,c}^k, ~\forall i,c,k$ in our experiments. This initialization renders the proximal operator an identical mapping, and thus PGD boils down to GD. Consequently, MetaProx exhibits no impact on the backbone method in the initial stage. As meta-training progresses, the PLFs dynamically adjust its parameter $\psi_{i,c}^k$ to learn a data-driven prior.
> Additionally, we also observed that the training of the PLFs occasionally diverge with the Adam optimizer, but remains robust when trained by SGD with Nesterov momentum. Numerically, this divergence appears linked to the instability in second moment estimation of the gradient.
>
> **5. Response to Question 4**
> The proposed MetaProx is expected to be particularly effective on more challenging tasks, which naturally necessitate sophisticated priors. As an example, in the ablation test (cf. Table 2), MetaProx improves MAML by $(53.58 - 48.70)/48.70 = 10.0\\%$ in the 5-way 1-shot setup, while the relative performance improvement is $11.3\\%$ in the 10-way 1-shot case.
>
> **6. Response to Question 5**
> The tradeoff is hidden in the regularization term of (1b). Consider for instance an isotropic Gaussian prior pdf $p(\boldsymbol{\theta_t}; \boldsymbol{\theta}) = \mathcal{N}(\boldsymbol{\mu}, \lambda^{-1} \mathbf{I}_d), ~\boldsymbol{\theta}:=[\boldsymbol{\mu}^\top, \lambda]^\top$, the corresponding regularizer can be analytically written as $\mathcal{R}(\boldsymbol{\theta_t}; \boldsymbol{\theta}) = -\log p(\boldsymbol{\theta_t}; \boldsymbol{\theta}) = \frac{\lambda}{2} || \boldsymbol{\theta}_t - \boldsymbol{\mu} ||_2^2 + \text{constant}$. In this example, the isotropic precision $\lambda$ serves as the (learnable) tradeoff weight between loss and regularization.

---

> > ### Comment · Reviewer_W1mq · 2023-11-13
> >
> > I appreciate the authors' detailed responses to my questions and the additional experiments which will be provided in later manuscript. My major concerns have been addressed. Looking forward to reviewing the revised manuscript.

---

### Official Review · Reviewer_Da9s · 2023-10-31

**Soundness:** 3 good
**Presentation:** 3 good
**Contribution:** 3 good
**Rating:** 6
**Confidence:** 4

**Summary:**

This paper presents a meta learning algorithm called MetaProx. When addressing meta learning algorithm with so-called proximal gradient descent, a central idea of this paper is to apply algorithmic unrolling. Algorithmic unrolling solves iterative optimization method by learning a deep neural network by cascading the steps of the iterative process. By applying algorithmic unrolling to proximal gradient descent, the paper attempts to enhance expressiveness and interpretability of the prior. Piecewise linear functions are chosen as proximal operators, for which, the paper provides error bounds theortically. In few-shot learning benchmarks, the paper shows advantages of the proposed method.

**Strengths:**

I noted the following strenghts:

1. The idea of bringing algorithmic unrolling to proximal gradient descent is new and interesting.

2. The paper derives theoretic error bounds for both smooth and non-smooth proximal operator, which makes the approach theoretically sound with certain level of technical depth.

3. The proposed method seems to perform well in the few shot benchmark. Improvements are consistent.

**Weaknesses:**

I thought following points could improve the paper.

1. I wonder if the dervied theoretic bounds can be validated empirically and the results appear in appendix.

In toy settings, it is often possible to compute the error bounds. Numerical examination could validate the bounds and further show how tight the error bound can be. Specifically, the paper claims that the derived bounds are tight. It could help if this claim is verified through experiments.

2. There are rooms to improve the presentation.

Introduction: Only few sentences are devoted to describe the contribution of the paper, while four paragraphs are devoted to motivation for this work. Within the descriptions of the contributions, most of the words are spent in giving benefits of the approach. I think this should appear after explaining what the actual approach is, e.g., explain what algorithm unrolling technique is, etc.

Related work: The paper misses related work section. Therefore, it was difficult to locate the contributions of the paper within the state of the art.

Section 3.1: I had to read other papers to understand algorithmic unrolling. Looking back, the provided explanations seem rather difficult and this paragraph might be throughly revised.

- The part on explainability may not be clear.

Yes, algorithmic unrolling can add interpretability by examining each layer mapping. But, the results look rather vague and I wasnt sure how this explainable prior can be useful in practice.

----------------------------------------------

Overall, the paper has many ideas and I am leaning towards recommending accept. I hope that my concerns are addressed during rebuttal.

**Questions:**

Questions and suggestions have been addressed in the section above.

---

> ### Author Response · Authors · 2023-11-11
>
> Thank you for your interest in this work and for providing the insightful suggestions. Next, the issues raised are addressed one-by-one below.
>
> **1. Regarding Weakness 1**
> As suggested by the reviewer, a toy numerical test will be provided next to demonstrate the tightness of the bound in Theorem 3.2. Consider tasks defined by linear relationship $y_t^n =\mathbf{w}_t^{\*\top} \mathbf{x}_t^n + e_t^n$, and linear prediction model $\hat{y}_t^n = f(\mathbf{x}_t^n;\boldsymbol{\theta}_t) := \boldsymbol{\theta}_t^\top \mathbf{x}_t^n$ with squared $\ell_2$ loss $\mathcal{L} (\boldsymbol{\theta}_t; \mathcal{D}_t^\text{trn}) := \frac{1}{2} \sum_n || y_t^n - \boldsymbol{\theta}_t^\top{\bf x}_t^n ||_2^2$, where the unknown oracle $\mathbf{w}_t^* \sim \text{Uniform}([-3,3]^d)$ and $\mathbf{x}_t^n \sim {\cal N}({\bf0}_d,\mathbf{I}_d)$. In the test, we set $K = 5, \alpha=0.01, d=64, |\mathcal{D}_t^\text{trn}|=8, \boldsymbol{\theta}^\text{init}=\mathbf{0}_d, A = 3$ and $C$ varying from 2 to 20. The target optimal proximal operator to be approximated is defined in eq.(27) with $G_1 = 1$. Additionally, the Lipschitz constant $G_2$ of $\nabla \mathcal{L}(\boldsymbol{\theta}_t; \mathcal{D}_t^\mathrm{trn})$ is numerically computed from the generated data $\\{ \mathbf{x}_t^n \\}_n$.
>
> The plot comparing the numerical PGD error and its upper bound can be found via this [anonymous URL](https://docs.google.com/presentation/d/e/2PACX-1vTtYYSMKjOnX8nlyNFWe3Q0980cG2MsoxkTPwFRY2joTSxkD4DwiQaxZFyZqYwwWWVu8AJao0r7pmWH/pub?start=false&loop=false&delayms=3000). It is observed that the numerical error aligns with the theoretical bound up to a small constant in the log-scale. This discrepancy arises because the upper bound considers the worst-case scenario, where the largest error between ${\rm prox}_{{\cal R}^*, \alpha}$ and ${\rm prox}^k$ is reached at each PGD step. Furthermore, this constant gap suggests that the numerical error is in the  same order with the bound (notice that both axes are in log-scale); that is, ${\cal O}(1/C^2)$. This empirically corroborates our theoretical proofs.
>
> **2. Regarding Weakness 2**
> We are in the process of revising the introduction to improve its clarity. In particular, an additional paragraph has been added to offer a comprehensive description of our methodology, before listing the contributions. Owing to space limitation, related work and an introduction to algorithm unrolling have been provided in the Appendices K and J, respectively.
>
> **3. Regarding Weakness 3**
> In practice, the visualization of the PLFs can be utilized to examine the impact of the prior when updating model parameters, thus providing guidance for practical model training process. In Figure 4, the acquired PLFs keep shallow layer weights being sparse around the initial value $\boldsymbol{\theta}^{\rm init}$ (that is, less updated) when $k$ is small, while deep layers can be updated freely along its gradient directions. This suggests, when fine-tuning a pre-trained large-scale model on a specific task, it is advisable to freeze the weights of the embedding function and exclusively train the last few layers with a relatively large step size in the initial epochs. Once these deep layers have attained sufficiently training, one can then gradually unfreeze the shallow layers and proceed with fine-tuning the entire model. In fact, this learned update strategy closely aligns with the widely adopted “gradual unfreezing” training approach for fine-tuning large-scale models, which has been proven effective in various practical applications of [computer vision](https://www.tensorflow.org/guide/keras/transfer_learning) and [natural language processing](https://aclanthology.org/P18-1031.pdf).

---

### Official Review · Reviewer_5m7L · 2023-11-01

**Soundness:** 3 good
**Presentation:** 3 good
**Contribution:** 3 good
**Rating:** 6
**Confidence:** 3

**Summary:**

The authors present an approach for meta-learning that estimates per-task model
parameters with an unrolled neural network. The neural network is derived from
proximal gradient descent on a regularized ERM problem (per-task); the
regularization term is learned by representing its proximal operator, which is
parameterized as a piecewise linear function that separates over its
coordinates. A theorem is given that bounds approximation error between
piecewise linear operators of this type and general $C^1$ maps asymptotically in
terms of the number of discretization points. Experimental results demonstrate
improved performance over certain meta-learning benchmarks on few-shot image
classification, and visualize the proximal operators learned in these tasks.

**Strengths:**

- The paper presents a clear and robust overview of background and prior work on
  meta-learning, making the authors' contributions easy to understand and
  appreciate in-context.

- The paper features a mixture of theory, justifying the approach, and
  experiment.

- Experimental results demonstrate the approach performs favorably on few-shot
  image classification with standard meta-learning datasets against standard
  benchmarks.

- The results on learned nonlinearities in section 4.4 are an interesting
  consequence of the authors' method. In interpreting these results, it would be
  helpful to know how the $\zeta$ parameters are initialized.

**Weaknesses:**

- The conceptual contribution is not exactly made clear. Maybe the clearest
  manifestation of this is that the presentation of the
  methodology does not make it clear why the authors' approach should be
  superior to other approaches to meta learning (say, to solving the formulation
  in equation (1)). The methodological discussion emphasizes that several
  commonly-used meta learning priors (under a common assumption on the "optimal
  regularizer") can be associated to proximal operators that can be represented
  as coordinatewise-applied piecewise linear functions. However, it does not
  connect back to specific tasks of interest in practice mentioned in the intro
  (robotics, say) and argue that the parametrized piecewise-linear proximal
  operators (essentially learnable nonlinearities in a neural network that
  implements the base-learner) are useful given specific structures the model
  parameters $\boldsymbol{\theta}_t$ have in these tasks. I think this is
  important to be able to support the authors claims of "expressivity" and
  "interpretability" made in (i) of the claimed contributions.

- Labeling the algorithm "MetaProx" seems like it might be a slight misnomer --
  based on the discussion on page 5 (after Assumption 3.1), it seems that what
  the authors are implementing is rather an amalgamation of one-dimensional
  picewise linear functions, which need not correspond to the prox of any
  function. In one dimension, it should be sufficient and necessary here to
  constrain the piecewise linear function to be monotone.

- It is not completely clear how the authors' eventual MetaProx algorithm
  differs from other meta learning approaches; most of the authors' discussion
  emphasizing the novelty of their approach seems to focus on contrasting with
  prior works on algorithm unrolling. For example, is this the first work that
  has made a connection between the problem in equation (1) and proximal
  gradient descent? A small section discussing this after the method has been
  presented would be helpful.


### Minor Issues

- Page 3: "...the optimization-based approaches solve (2b) ..." -- should this
  be (1b)? The rest of this setence discusses $\mathcal{R}$, which does not
  appear in equation (2).
- Text before equation (11) on page 7: seems to be a dangling clause, "...has
  gained attention in various PGD-guided a [sic?]."
- Notation for continuously differentiable functions: as I understand it, using
  $\mathbb{C}^k$ for this class is nonstandard (and clashes with the usual
  notation for $k$-tuples of complex numbers in a jarring way). I guess this is
  an attempt to avoid a conflict with the parameter $C$ in the authors'
  algorithm; what about using $\mathcal{C}^k$ or $\mathsf{C}^k$ for this class?
- Is it correct that Theorem 3.2 involves Assumption 3.1? If so, it would be
  good style to reference this in the theorem statement.

**Questions:**

- The authors motivate their method at the start of section 3.1 by contrasting
  their method, which learns a prior over model parameters
  $\boldsymbol{\theta}_t$, with prior work on algorithm unrolling, where the focus
  is to learn a prior over signals $\mathbf{x}_t$ for one fixed task. It would
  be helpful in supporting this point if the authors presented a concrete
  example of a class of tasks where this difference can be appreciated. For
  example, what form does this distinction take in a family of Gaussian
  denoising tasks -- a typical setting for algorithm unrolling? I think this
  would be helpful, as the presentation of the computational methodology in the
  rest of the section tends towards abstraction.

- The notion of an "optimal" regularizer $\mathcal{R}^*$ is introduced on page
  4, and plays a role in the authors' methodology. In what sense is this
  regularizer "optimal"? I could not find any discussion of this notion in the
  background section.

- Is the content of Theorem 3.2 mainly an application of results on
  one-dimensional piecewise linear function approximation (with uniform
  discretization) of $C^1$ functions?

- What is the "apples-to-apples" comparison mentioned when discussing the
  experimental results on page 8? Are there SotA meta-learning methods that have
  been excluded from this table? It seems natural that the MetaProx approach
  might perform better than methods that use less computation, since Algorithm 2
  could be interpreted for natural losses $\mathcal{L}$ as a multilayer residual
  network (with some per-task conditioning) with learnable nonlinearities, if
  I'm not mistaken.

---

> ### Author Response · Authors · 2023-11-11
>
> Thanks for the thorough review of our manuscript and the valuable feedback provided. The issues raised have been carefully addressed one-by-one as follows.
>
> **1. Response to Weakness 1 & Question 1: example of claimed superior expressiveness.**
> As a straightforward yet illuminating example, consider regression tasks defined by the 1-d linear data model $y_t^n = w_t^* x_t^n + e_t^n$, where the unknown per-task weights $\\{w_t^*\\}_{t=1}^T$ are i.i.d. samples from the oracle pdf $p(w^*):=\frac{1}{2}\text{Uniform}(-11,-10)+\frac{1}{2}\text{Uniform}(10,11)$, and $e_t^n\sim{\cal N}(0,\sigma_e^2),\forall t$ is the additive white Gaussian noise. Further, consider a linear prediction model $\hat{y}_t^n=f(x_t^n;\theta_t):=\theta_tx_t^n$. Since $p(w^*)$ is symmetric, the optimal Gaussian prior of $\theta_t$ for this case must have a mean of 0. In other words, if the prior pdf is chosen to be Gaussian $p(\theta_t;\theta)={\cal N}(\mu,\lambda^{-1}),\theta:=[\mu,\lambda]^\top$, the optimal parameter $\theta^*$ must consist of $\mu^*=0$ and some $\lambda^* \in \mathbb{R}$. As a result, the corresponding regularizer for this prior is ${\cal R}(\theta_t;\theta^*)=\frac{\lambda^*}{2}\theta_t^2$, which prevents $\theta_t$ deviating far from $0$. However, the ground-truth task model implies that the optimal $\theta_t^*$ should belong to the set $S:=[-11,-10]\cup[10,11]$, and the regularizer is thus a barrier for optimizing $\theta_t$. In contrast, if the prior pdf is allowed to be non-Gaussian, the optimal prior will be the ground-truth one, i.e. $p(\theta_t; \theta^*)={\rm Uniform}(S)$. The corresponding proximal operator in this case is the projection $\text{prox}(z)=\mathbb{P}_S(z)$, which is exactly a piecewise linear function (PLF) driving $\theta_t$ to the oracle set. In summary, the Gaussian pdf fails to match the underlying prior due to its inherent unimodality, while the more expressive PLF-induced prior can perfectly align with the groundtruth prior to enhance the task-level learning.
> In practical tasks such as drug discovery and robotic manipulations, the ground-truth model parameters can exhibit similar multi-modal pdf defined on a bounded set. In drug discovery for instance, the efficacy of a drug might only manifest when one component accounts for a specific portion.
> To enhance the clarity of the paper, this example and a corresponding experiment have been included in Appendix I of the revised manuscript. We express our gratitude for the insightful suggestion.
>
> **2. Response to Weakness 2: misnomer and monotonicity of PLF.**
> In fact, a sufficient condition for the proximal operator to be monotone is the convexity of the regularizer $\cal R$. Specifically, denoting by $x=\text{prox}_{{\cal R},\alpha}(z)$, it follows from stationary point condition that $x\in z-\alpha\partial{\cal R}(x)$, which can be non-monotone when $\cal R$ is non-convex. In addition, the PLF in this case still induces a valid regularizer, although it may not have a closed-form solution. Please check Remark C.3 in Appendix C for more details.
>
> **3. Response to Weakness 3: comparison between MetaProx with other meta-learning algorithms.**
> Existing meta-learning algorithms rely on either a *blackbox NN* or a *preselected pdf* (such as a Gaussian one) to parameterize the prior. However, the NN often lacks *interpretability* and the chosen pdf can have *limited expressiveness*; that is, it may have insufficient ability to offer an accurate fit. Consider for instance a preselected Gaussian prior pdf, which is inherently unimodal, symmetric, log-concave, and infinitely differentiable by definition. Such a prior may not be well-suited for tasks with multimodal or asymmetric parametric pdfs, as illustrated in the aforementioned example. In contrast, our MetaProx introduces *data-driven prior* (induced by the learnable PLFs) which dynamically adjust its form to fit the provided tasks and offers the desired interpretability. Our theory and numerical tests are also centered around this claim of prior expressiveness and interpretability.
> In response to the suggestion from the reviewer, we have incorporated this comparison into Sec.3 of the revised manuscript.
>
> **4. Response to Minor Issues.**
> We extend our appreciation for pointing out the typos and improper notations in the manuscript, which will be corrected accordingly. Regarding the last minor issue, a more formal version of Theorem 3.2 and 3.3 can be found in Appendices A and B, with Assumption 3.1 clearly involved in the statement.
>
> **5. Response to Question 2: definition of the optimal regularizer.**
> The optimal regularizer is defined to be the function $\cal R$ (possibly restricted to a space such as ${\cal C}^0$ or ${\cal C}^1$) that minimizes the meta-learning objective (1a). This missing definition has been added to the revised manuscript.

---

> ### Author Response · Authors · 2023-11-11
>
> **6. Response to Question 3: connection between Theorem 3.2 and 1-d PLF approximation.**
> The proof of Theorem 3.2 relies on the 1-d PLF approximation; cf. Lemmas A.5 and B.1. However, the proof is not a trivial application of the PLF approximation, particularly in the construction of the example for tightness; see Theorems A.6 and A.7 for more details.
>
> **7. Response to Question 4: elaboration of “apples-to-apples” comparison and the extra computational cost of MetaProx.**
> Here “apples-to-apples” comparison ensures the methods listed in the tables are evaluated using the same task-specific learning model. In other words, the dimension $d$ of model parameter $\theta_t$ and the model architecture are consistent among the methods. For fairness, results obtained with larger models have been excluded from the table. Moreover, complexity analysis has been provided in Appendix F both theoretically and numerically, revealing that the learning of PLFs contributes only marginally ($< 5\\%$) to the overall complexity compared to its backbone. As a result, our MetaProx can serve as a flexible plug-in module to enhance the performance of backbone meta-learning algorithms without markedly adding to its complexity. Notably, this benefit is realized through the acquisition of a more expressive prior.

---

> ### Comment · Reviewer_5m7L · 2023-11-22
> **thanks**
>
> Dear authors,
>
> Thank you for the thorough response to my review. I am happy with the clarifications you have provided, and will increase my score. I found the intuitive example you provided (first point response) to be helpful in clarifying the utility of your method.
>
> I want to push back slightly on the convexity and prox issue, though -- I am not sure this is correct. An excellent characterization of this issue has been provided recently by Gribonval and Nikolova in https://centralesupelec.hal.science/hal-01835101/. One sees (see Theorem 1) that the characterization of when a map is a proximal operator amounts to it being the gradient of a convex function. For a 1d example, and in the presence of differentiability, I believe this is equivalent to the map in question being monotone (for example, imagine the twice-differentiable case, and note that the map is equal to the integral of its second derivative (up to a constant) -- and by convexity, this second derivative is positive).
>
> I believe the issue you have in mind about convexity of the regularizer is different -- Gribonval and Nikolova clarify that the classical Moreau's characterization of proximal operators is that the map is the gradient of a convex function *and nonexpansive*. The characterization I discussed above (Theorem 1 of Gribonval and Nikolova) shows that in the nonconvex case, it is necessary and sufficient to discard this nonexpansivity.
>
> This issue is not affecting my rating of the paper, but I do hope we can come to an agreement on this and clarify it in the work.

---

> > ### Author Response · Authors · 2023-11-22
> > **Follow-up response**
> >
> > Thank you for your timely response, and for raising the score. We agree that monotonicity is a crucial characteristic for the PLF to qualify as a valid proximal operator. In practice, the sought monotonicity can be established by enforcing $\psi_{i,c}^k \le \psi_{i,c'}^k,~\forall c\le c'$. And interestingly, we have observed that the learned PLFs are exactly monotone functions, even without an explicit constraint. This observation suggests an inherent preference for monotonic PLFs by the data.
> >
> > This valuable insight will be incorporated into Appendix C of our manuscript.

---

> > > ### Comment · Reviewer_5m7L · 2023-11-22
> > > **thanks**
> > >
> > > Yes, I agree re: enforcing and this being straightforward. And it is interesting to hear the empirical observation. Thank you for incorporating this into the revision.

---

### Public Comment · ~Weisen_Jiang1 · 2023-11-16
**MetaProx has been named as a meta-learning algorithm proposed at NeurIPS 2021**

Dear authors,

Congrats on this meta-learning work.

Note that the name **MetaProx** has been used in our paper (Effective Meta-Regularization by Kernelized Proximal Regularization
https://proceedings.neurips.cc/paper/2021/hash/dcc5c249e15c211f21e1da0f3ba66169-Abstract.html, **NeurIPS 2021**).
Our work proposes a **meta-learning algorithm for learning a prior model**. We sincerely ask the authors to rename their algorithm to avoid confusion.
Also, we believe discussing their differences in the paper is also necessary.

Best regards,

---

> ### Author Response · Authors · 2023-11-16
>
> Hi Weisen,
>
> To avoid [potential confusion](https://scholar.google.com/scholar?hl=en&as_sdt=0%2C24&q=MetaProx&btnG=), the name of our method will be changed to `MetaProxNet` in our revised manuscript. In addition, the comparison will be included in the Appendix.
>
> Thanks,
> Authors of paper #4443

---

> > ### Public Comment · ~Weisen_Jiang1 · 2024-03-20
> > **lack of discussion**
> >
> > Dear authors,
> >
> > For the current camera-ready version, the related work `MetaProx` is missing. I think it is necessary to discuss the **difference between MetaProxNet and MetaProx** in the paper.
> >
> > 1. their names are very similar
> > 2. both of them use proximal operation
> >
> > Best,

---

### Comment · Area_Chair_LHGa · 2023-11-22

Dear reviewers,

This a reminder that deadline of author/reviewer discussion is AOE Nov 22nd (today). Please engage in the discussion, check if your concerns are addressed, and make potential adjustments to the rating and reviews.

Thank you!
AC

---

### Meta-Review · Area_Chair_LHGa · 2023-12-04

**Metareview:**

This work proposes a new approach using unrolled proximal neural networks to learn and represent priors in a more expressive and interpretable way. Both the theoretical and empirical results look very promising and interesting. The reviewers reached consensus to accept this paper, given its sound and solid technical contributions. Please incorporate the reviewers' feedback in the revision including clarity of the method, section 3.1 and promised new results for Caltech-UCSD Birds (CUB)-200-2011 dataset and Meta-Dataset. I recommend to accept this paper.

**Justification For Why Not Higher Score:**

There are still some issues with clarity and inclusion of new results promised by the authors which are difficult to evaluate. These are, however, relatively minor.

**Justification For Why Not Lower Score:**

This work proposes a new approach using unrolled proximal neural networks to learn and represent priors in a more expressive and interpretable way. Both the theoretical and empirical results look very promising and interesting. The reviewers reached consensus to accept this paper, given its sound and solid technical contributions.

---

### Decision · Program_Chairs · 2024-01-16

Accept (poster)